# Ecological diversification of sea catfishes is accompanied by genome-wide signatures of positive selection

Melissa Rincon-Sandoval [1,10], Rishi De-Kayne[2], Stephen D. Shank[3], Stacy Pirro[4], Alfred Ko'ou[5], Linelle Abueg [6], Alan Tracey[6], Jackie Mountcastle[6], Brian O'Toole[6], Jennifer Balacco[6], Giulio Formenti [6], Erich D. Jarvis [6], Dahiana Arcila [7], Sergei L. Kosakovsky Pond[3], Aaron Davis [8], Devin D. Bloom[9] & Ricardo Betancur-R [7,10] ✉

Habitat transitions have shaped the evolutionary trajectory of many clades. Sea catfishes (Ariidae) have repeatedly undergone ecological transitions, including colonizing freshwaters from marine environments, leading to an adaptive radiation in Australia and New Guinea alongside non-radiating freshwater lineages elsewhere. Here, we generate and analyze one long-read reference genome and 66 short-read whole genome assemblies, in conjunction with genomic data for 54 additional species. We investigate how three major ecological transitions have shaped genomic variation among ariids over their ~ 50 million-year evolutionary history. Our results show that relatively younger freshwater lineages exhibit a higher incidence of positive selection than their more ancient marine counterparts. They also display a larger disparity in body shapes, a trend that correlates with a heightened occurrence of positive selection on genes associated with body size and elongation. Although positive selection in the Australia and New Guinea radiation does not stand out compared to non-radiating lineages overall, selection across the prolactin gene family during the marine-to-freshwater transition suggests that strong osmoregulatory adaptations may have facilitated their colonization and radiation. Our findings underscore the significant role of selection in shaping the genome and organismal traits in response to habitat shifts across macroevolutionary scales.

Understanding how different lineages have adapted to new environments over macroevolutionary time scales, and how these habitat transitions have shaped contemporary patterns of biodiversity is a major pursuit in biology. Many studies investigating adaptation and diversification focus either on model study systems with few ecological shifts, allowing researchers to delve into specific axes of adaptation, or on complex adaptive radiations, where adaptation along multiple axes has typically ocurred[1–3]. While each approach has its benefits, such as the ability to control for variables in small model study systems[4,5] and the representation of complex multidimensional

adaptation in adaptive radiations, it is often challenging to draw broadly applicable conclusions from these investigations. The ability to generate and analyze high-quality genomic data and reference genomes even for non-model clades now facilitates investigations into genomic changes that may be associated with the colonization of new habitats across broad phylogenetic scales[6,7]. Across multiple clades, habitat transitions have been shown to drive adaptive radiation via ecological opportunity[8–10]. By integrating genomic data with ecological and evolutionary analyses, we can gain valuable insights into the mechanisms underlying adaptation and unravel the genomic basis of

evolutionary changes in response to ecological shifts both in adaptive radiations and more generally across the tree of life.

Transitions between marine and freshwater habitats comprise a major axis of ecological diversification in aquatic organisms[11]. Marine lineages commonly undergo major shifts in their selection regimes when they enter and diversify into freshwater habitats[12–14]. Marine-to-freshwater transitions have occurred repeatedly in several animal groups, including annelids[11], mollusks[15], arthropods[16], and teleost fishes[17]. These ecological shifts typically involve euryhaline groups that can tolerate a wide range of salinity and thus are able to overcome the physiological stress imposed by the new osmotic environment[18]. Among fishes, many clades include euryhaline species, but only a subset of these groups have undergone permanent habitat transitions between marine and freshwater ecosystems, and few of these transitions have resulted in adaptive radiations[19,20]. Given the hyper diversity of freshwater fishes (~40% of all fish species) relative to the tiny fraction (<0.01%) of the overall aquatic realm represented by freshwater[21,22], it is clear that rivers and lakes provide a wealth of ecological resources and potential opportunities for diversification[23].

Recent studies have identified numerous genomic regions and genes that are thought to play key roles in the successful colonization of freshwater environments by marine-derived freshwater and euryhaline fishes, which can tolerate a wide range of salinity levels[5,24]. For example, genomic analyses of the euryhaline European sea bass identified expansions of five gene families (aquaporins, claudins, arginin-vasotocin or *AVT* receptors, prolactin or *PRL* and its receptor *PRLR*) that are associated with ion and water regulation, a putative adaptation to variation in salinity[25]. Another recent study[26] showed that multiple duplications of the fatty acid desaturase gene or *FADS2* (a metabolic gene) took place independently in ray-finned fish lineages that colonized and radiated in freshwater habitats, but not in their marine-only relatives, suggesting that *FADS2* played a key role in overcoming nutritional constraints associated with freshwater colonization[26]. At least six different gene families (i.e., transporter proteins such as aquaporins, ion channels, osmoregulatory hormones and their receptors, metabolic genes, immune genes, and heat shock proteins) have been associated with the ability of marine-derived lineages to successfully colonize and diversify in freshwaters[5,24,27–30]. However, the universality of these genes in helping other species colonize or adapt to freshwater environments is unclear[31].

Geographic regions that provide the ecological opportunity for repeated, evolutionary-independent transitions of marine lineages to freshwater habitats constitute excellent systems to investigate patterns of adaptation to new environments. These systems allow the investigation of key aspects regarding adaptation to new environments, including the identification of abiotic and biotic factors that may drive or constrain a lineage's transition to a new habitat, and the repeatability of evolution, specifically whether adaptation during habitat transitions occurs in similar ways each time it evolves[32]. Tropical rivers in Australia and New Guinea (AU-NG) represent an exceptional setting for such studies. Rivers in the AU-NG region are depauperate of primary freshwater fish forms—i.e., salt intolerant groups, such as carps, minnows, characins, knifefishes, and most catfishes[33]—that are otherwise important components of nearly all other continental masses. In AU-NG, the majority of freshwater fish lineages consist of tropical groups that originated in marine environments, encompassing 28 families and ~475 species, with at least seven of these lineages exhibiting adaptive radiations[34–37]. Thus, the marine-derived freshwater fishes in AU-NG provide exceptional case studies for investigating habitat transitions, the broader adaptive radiation process, and the genomic signatures that underlie these radiations and are associated with ecological adaptation and speciation[13].

A prominent example of a marine-derived freshwater fish radiation in the AU-NG region is the sea catfishes in the family Ariidae[13,38]. Ariids have a global distribution, inhabiting warm-temperate and tropical regions, and exhibit a range of habitat preferences across marine to freshwater environments. Phylogenetic comparative analyses showed ariids underwent an initial transition from freshwater habitats--the ancestral habitat condition for catfishes--to marine environments, followed by multiple instances of recolonization of freshwater habitats in different geographical regions where the group is distributed[13,38]. While most of these recolonizations have resulted in non-radiating clades, colonization of freshwaters in AU-NG have spurred an adaptive radiation[13,34]. Freshwater ariids tend to exhibit lower rates of lineage diversification and morphological evolution compared to their marine counterparts (Fig. 1e, f); however, the opposite holds true for freshwater lineages in AU-NG. Within AU-NG, ariid lineages accumulated rapidly during the early history of the clade, followed by a gradual decline (Fig. 1g[13])—a characteristic pattern indicative of Simpsonian adaptive radiation. Freshwater species in this region are distinguished by their high levels of trophic diversity and dietary partitioning, including filter feeding, plant feeding, frugivory, piscivory, lepidophagy, and insectivory[13,39]. This independent and recurrent colonization of freshwaters by marine ariid lineages is thought to have been primarily influenced by open niche space, which transitioning fish species exploited for establishment and diversification, most notably in AU-NG[13].

In addition to transitions between marine and freshwater habitats or variations in salinity tolerance (from a narrow range in stenohaline species, to a broad tolerance in euryhaline species), shifts along the benthic-pelagic axis have also played a significant role in driving divergence within adaptive radiations of fishes. These shifts are often associated with changes in body shape and craniofacial structure, which are linked to resource utilization and diet preferences[3,5,40]. Well-known examples of benthic-pelagic divergences include model clades (e.g., stickleback[41], cichlids[42], whitefish[43]) that have repeatedly evolved from deep-bodied benthic forms with truncated caudal fins into slender midwater species with forked caudal fins[44]. While many of these clades have diversified relatively recently, similar mechanisms and constraints to evolution are thought to have occurred at deeper evolutionary scales in other groups[44]. One aspect of craniofacial morphology that is often linked to transitions in the water column, diet composition, and prey size is the development of gill rakers, which are dermal bones critical for food acquisition[3,27,40]. Species with a higher number of gill rakers possess enhanced feeding efficiency on zooplankton in pelagic habitats[40], while those with lower raker counts are typically associated with foraging in benthic environments[40]. Notably, the AU-NG radiation contains the majority of pelagic planktivore species with high raker counts in ariids. As a result ariid clades that have independently transitioned into pelagic planktivory provide an opportunity to investigate whether genes known to be associated with body shape variation[5,29,30] (e.g., *Cx43*, *MMP9*, *SEMA3D*, *BMP4*, *LBH*, *PTCH1*, *ACTB1*, *RPS18*, *HSP90A*, *RPS11*, *TBP*, *HPRT1*, *GAPDH*) and gill raker morphology[3,27] (e.g., *EDA*, *EDAR*, *FGF20A*) in other clades have also played an important role in facilitating the diversification along the benthic-pelagic axis in Ariids.

Here, we generate a de novo chromosome-level genome and short-read assemblies for 66 ariid species, and integrate phylogenetic comparative methods with phylogenetic genotype-to-phenotype (PhyloG2P) approaches[45] to examine the genetic basis for trait evolution in a non-model clade and investigate the underlying factors associated with habitat shifts across macroevolutionary scales. More specifically, we examine signatures of selection, variation in transposable element diversity and repeat content along the genome, and phenotypic disparity, associated with three major ecological transitions: marine to freshwater, stenohaline to euryhaline, and transitions within the water column. We hypothesize that coding genes functionally associated with key ecological traits are common targets of selection during ecological transitions, resulting in correlated changes in trait variation across large phylogenetic scales. We also expect to

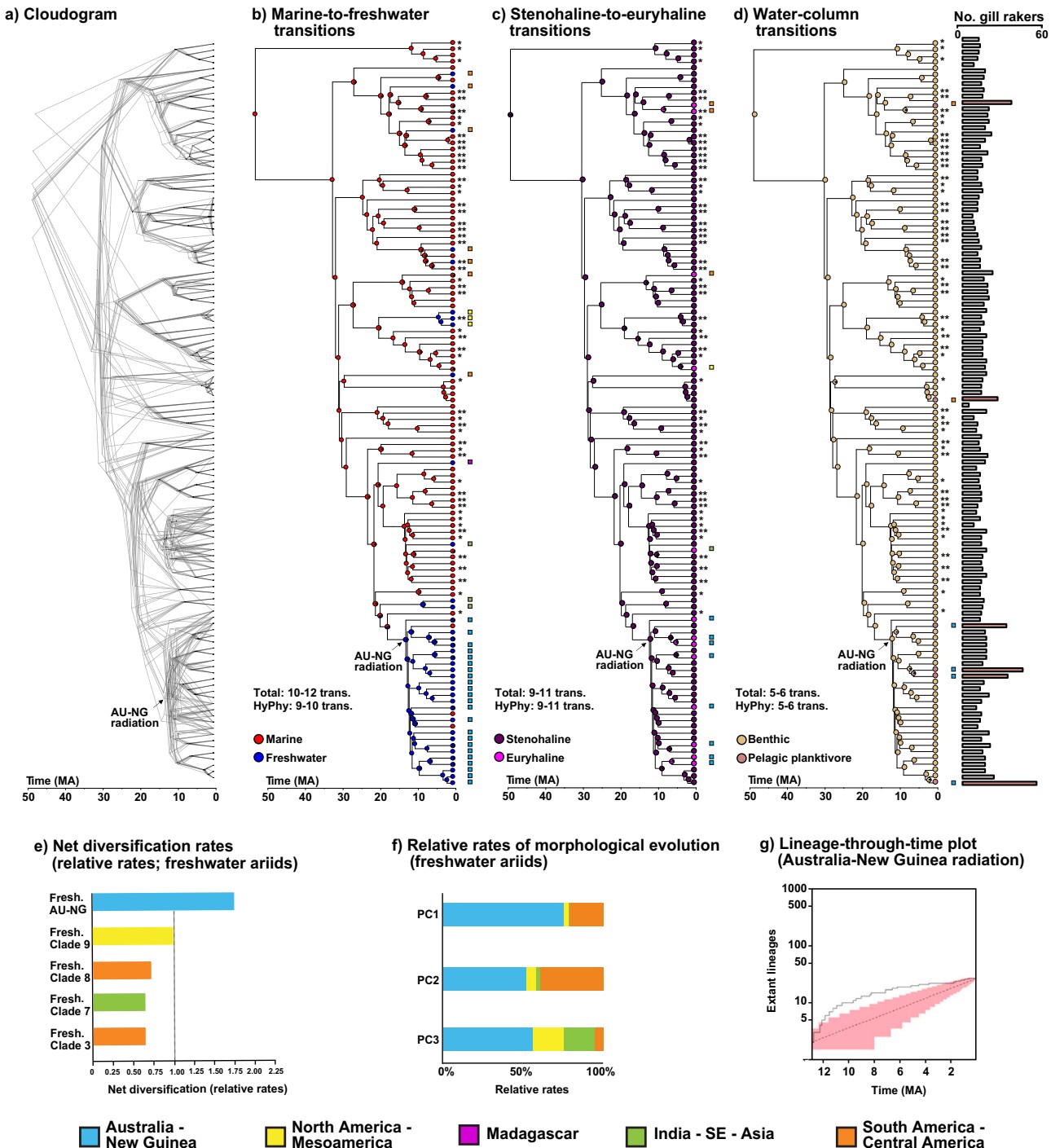

**Fig. 1 | Phylogenomic trees, ancestral habitat reconstructions, morphological evolutionary rates, and lineage through time plots for ariids. a** Cloudogram depicting the complete dataset and four subsets with 265 genes each, generated using concatenation ML (RAxML) or multi-species coalescent (ASTRAL-III) analyses. Time-calibrated topologies were estimated using 14 calibration points in MCMCTree. **b** SIMMAP-based reconstruction of marine-to-freshwater transitions, with marine (red) and freshwater (blue). **c** SIMMAP reconstruction of stenohaline-to-euryhaline transitions, represented by purple (stenohaline) and magenta (euryhaline) pies. **d** SIMMAP reconstruction of benthic-to-pelagic planktivore transitions, depicted with amaranth (benthic, few rakers) and pink (pelagic, many rakers) pies; bars indicate gill raker counts on first branchial arch. Trees b-d show the master tree topology (ASTRAL-III); see Figs. S10–S12 for reconstructions on all trees. Asterisks (*) denote tips sequenced via exon capture, while double asterisks (**) indicate tips placed based on legacy markers only. Color squares indicate geographic regions for freshwater, euryhaline, and pelagic species. Note that transitions in the opposite direction (e.g., freshwater to marine) also occurred but are not noted here; they are discussed in the main text. **e** Net diversification rates standardized against background rates in ariids. **f** Relative rates of morphological evolution in regional freshwater ariid assemblages. **g** Lineage through time (LTT) plot for the Australia-New Guinea (AU-NG) adaptive radiation, demonstrating early burst diversification. Plots e-g are adapted from Betancur-R. et al.[13].

identify changes in genomic repeat content associated with environmental shifts, given that recent research suggests that repeats may both underpin phenotypic differentiation itself through the modulation of gene expression, or that repeats may accumulate following the occupation of a new riche as selection is relaxed[46–48]. Lastly, a key focus of our investigation is the comparative analysis of patterns of selection between the marine-derived freshwater adaptive radiation in AU-NG, which encompasses multiple euryhaline and pelagic planktivore lineages, and analogous clades observed in other marine-derived freshwater, euryhaline, or pelagic planktivore lineages elsewhere. We thus expect to observe more pronounced signatures of gene selection and variations in repeat content in AU-NG relative to their non-radiating counterparts in other regions.

## Results and discussion

### Sequencing and assembly of reference and short read genomes for 66 ariid species

To examine ecological transitions spanning the evolutionary history of sea catfishes and assess whether these transitions are correlated with convergent phenotypic and genomic changes we employed phylogenetic comparative and genomic analyses within a robust macroevolutionary framework based on multiple phylogenomic trees (Supplementary Note 1). First, we generated a high-quality chromosome-level genome of the lesser salmon catfish from Australia (*Neoarius graeffei*), and short-read genome assemblies from 66 species (including *N. graeffei* and one outgroup; produced using ~30x coverage Illumina NovaSeq data). The high-quality reference genome for the lesser salmon catfish was produced through the Vertebrate Genome Project (VGP) and resulted in a 2.34 Gbp assembly comprising 372 contigs and 38 scaffolds. The assembly has a N50 scaffold length of 83,992 Mb, a BUSCO completeness of 97.8% (Actinopterygii OrthoDB; v5.4.3), and a total of 28 chromosomes (10 scaffolds remain unassigned; see *Supplementary Information*). Short-read genomes were pre-assembled using SPAdes v3.13.1 and scaffolded with RagTag v2.1.0 using guided information from the reference genome. RagTag assemblies have a N50 of 8.865–67.48 Mbp (mean 55.54 Mbp; *Supplementary Information*, Fig. S7b; Supplementary Data S4-S7), and a BUSCO completeness of between 73.2 and 99.6% (mean 81.63%; *Supplementary Information*, Fig. S7c; Supplementary Data S5).

### Phylogenomic time trees for sea catfishes consistently group clades by geography

We integrated whole-genome data with newly generated exon capture markers from an additional 18 species (1051 markers)[49] and legacy mitochondrial and nuclear markers previously sequenced for an additional 36 species[13], and use these data to produce robust phylogenomic inferences across the entire Ariidae clade. The resulting data matrix includes 1039 exon[49] and seven legacy markers, consisting of 242,949 nucleotide sites for 119 (out of ~157) ariid species plus an outgroup, with 76% data completeness (*Supplementary Information*, Supplementary Data S1). Phylogenomic analyses using maximum likelihood and coalescent-based approaches using the complete dataset ('master tree' hereafter) resulted in well-supported trees that were consistent with previous studies[13,38,50,51], providing a robust framework for further analyses.

All phylogenetic reconstructions consistently support the division of the family into two subfamilies (Ariinae and Galeichthyinae). Within Ariinae, the results identified seven major geographically clustered clades, including the AU-NG adaptive radiation. Divergence-time analyses using Bayesian approaches and 14 calibration points (*Supplementary Information*, Figs. S2, S9; Supplementary Table S1) estimate the crown age of Ariidae to be 48.3 Ma, and the AU-NG radiation to have occurred 13.0 Ma. We also assembled four largely independent gene subsets, each with 265 exon markers (*Supplementary Information*, Supplementary Data S2), which we subjected to the same analytical

approaches as the complete data matrix. These analyses collectively produced 10 alternative trees, which we used to accommodate phylogenetic and divergence time uncertainty in downstream comparative analyses. We also inferred three additional trees using up to 3,551 BUSCO genes obtained from the whole genomes assembled (see below), producing highly consistent topologies (*Supplementary Information*, Fig. S7). Detailed taxonomic relationships and age estimates for different trees are provided in the *Supplementary Information*, Supplementary Note 2.

### Ariids have repeatedly colonized freshwater environments and evolved euryhalinity and pelagic ecological strategies

To identify habitat transitions, we conducted three independent estimates of ancestral ecologies across the ariid phylogeny in SIMMAP using the assigned extant sampled species to major habitat groups across three axes of variation: (i) marine or freshwater; (ii) stenohaline or euryhaline; (iii) benthic or pelagic planktivore. These estimates accounted for uncertainty in tree topology and habitat occupancy (see *Supplementary Information*, Materials and Methods). For marine-to-freshwater transitions, our results are consistent with previous work[13,38], showing that marine habitats represent the most likely ancestral condition for ariids, with an estimated 10–12 transitions identified across different trees in various regions worldwide (Fig. 1; *Supplementary Information*, Fig. S10; Supplementary Note 3). Furthermore, these trees reveal that 2–3 lineages within the AU-NG adaptive radiation had reverted to the ancestral marine condition. We estimated 9-11 shifts towards euryhalinity in diverse biogeographic settings, with 5–7 of these transitions occurring specifically within the AU-NG adaptive radiation (Fig. 1; *Supplementary Information*, Fig. S11). Finally, for water-column transitions, we identified 5–6 transitions to pelagic planktivory (Fig. 1 and *Supplementary Information*, Fig. S12). Among these transitions, 3–4 took place within the AU-NG adaptive radiation, while two were found in marine lineages in South and Central America. Recurrent ecological transitions in ariids, characterized by independent shifts across clades, exemplify evolutionary replication and offer a unique system to study convergent and non-convergent genomic signatures in radiating and non-radiating lineages globally

We also examined the average proportional time spent in each habitat or state by ariid lineages. Considering the complete taxonomic sampling, ancestral lineages spent ~78% of their time as marine, 85% as stenohaline, and 95% as benthic (61%, 80% and 83%, respectively, considering trees with whole-genome species only that were used for positive selection analyses). In contrast, derived lineages spent on average, 22% of their time as freshwater, 15% as euryhaline, and 5% as pelagic planktivore (39%, 20% and 17%, respectively, for trees with whole-genome species only; Supplementary Data S9).

### Candidate genes noted to underpin fish adaptation are more prone to selection in connection with transitions compared to non-candidate genes

We compiled a set of 2,310 genes, including 249 candidate genes previously associated with fish adaptations[5,24,26,27,29,30,52–55], along with 2,061 non-candidate genes. Using various tools implemented in HyPhy v2.5.45 and later (aBSREL, BUSTED-E, BUSTED-PH, RELAX, and MEME; see Methods), we examined selection patterns associated with habitat transitions (marine-to-freshwater, salinity tolerance, and water-column) in Ariidae. After applying the Benjamini-Hochberg false discovery rate correction and other filters (BUSTED-E) to mitigate alignment or sequencing error, we retained 119 high confidence positively selected genes (PSGs) that we further tested across different transition schemes in Ariidae (see Methods and Fig. 2; *Supplementary Information*, Figs. S15–S18; additional details are given in *Supplementary Information*, Supplementary Materials and Methods, and Supplementary Note 4).

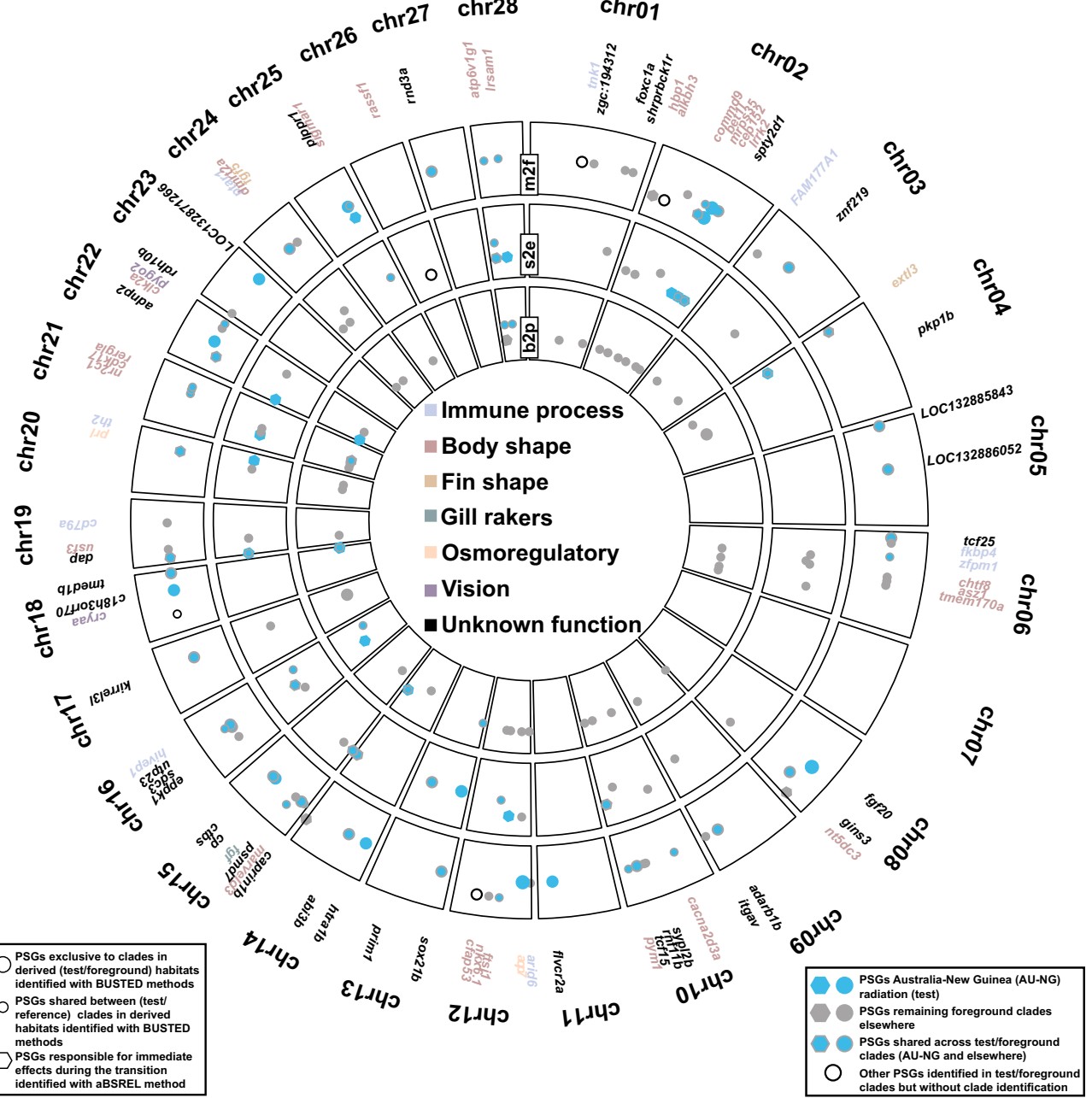

**Fig. 2 | Positively selected genes (PSGs) in association with habitat transitions relative to coordinates in the *Neoarius graeffei* genome.** PSGs with color-coded functional categories were identified by BUSTED-E, aBSREL, and BUSTED-PH for derived/foreground habitat states (freshwater, euryhaline, and pelagic planktivore). Each of the three-layered circles in the plot represents a habitat transition. m2f: marine-to-freshwater transitions; s2e: stenohaline-to-euryhaline transitions; b2p: benthic-to-pelagic planktivore transitions. Source data are provided as a Source Data file.

As expected, candidate genes showed a higher incidence of positive selection across all transitions compared to non-candidate genes (0.4-8% vs. 0.04-1.4%, respectively; see *Supplementary Information*, Fig. S14), confirming the importance of these candidate genes in ecological specialization. Overall, analyses of marine-to-freshwater transitions identified the highest number of genes under positive selection on the foreground lineages, followed by stenohaline-to-euryhaline and water-column transitions. However, this trend is partly influenced by the number of branches tested in each habitat transition (see below; Supplementary Data 1; *Supplementary Information*, Fig. S4). Across these transitions, genes related to ecological specialization (development, metabolism, immunity, reproduction, and transport) consistently experienced diversifying positive selection (see Fig. 2). Although positive selection is observed throughout the

genome, chromosomes 2, 10, and 12 in the reference exhibit a higher incidence, while chromosomes 7, 17, and 23 show just one or none of the positively selected genes (Fig. 2). We identified 21 biological process Gene Ontology (GO) terms across all transitions (*Supplementary Information*, Fig. S13; Supplementary Data S10), which were significantly overrepresented among PSGs across the radiation. These include terms associated with cellular processes, biological regulation, metabolism, immune response, stimulus response, and reproductive processes.

**Younger freshwater lineages reveal more positive selection than their ancestral marine counterparts**

For marine-to-freshwater transitions, we used two coding strategies to account for euryhaline species (including and excluding them) based

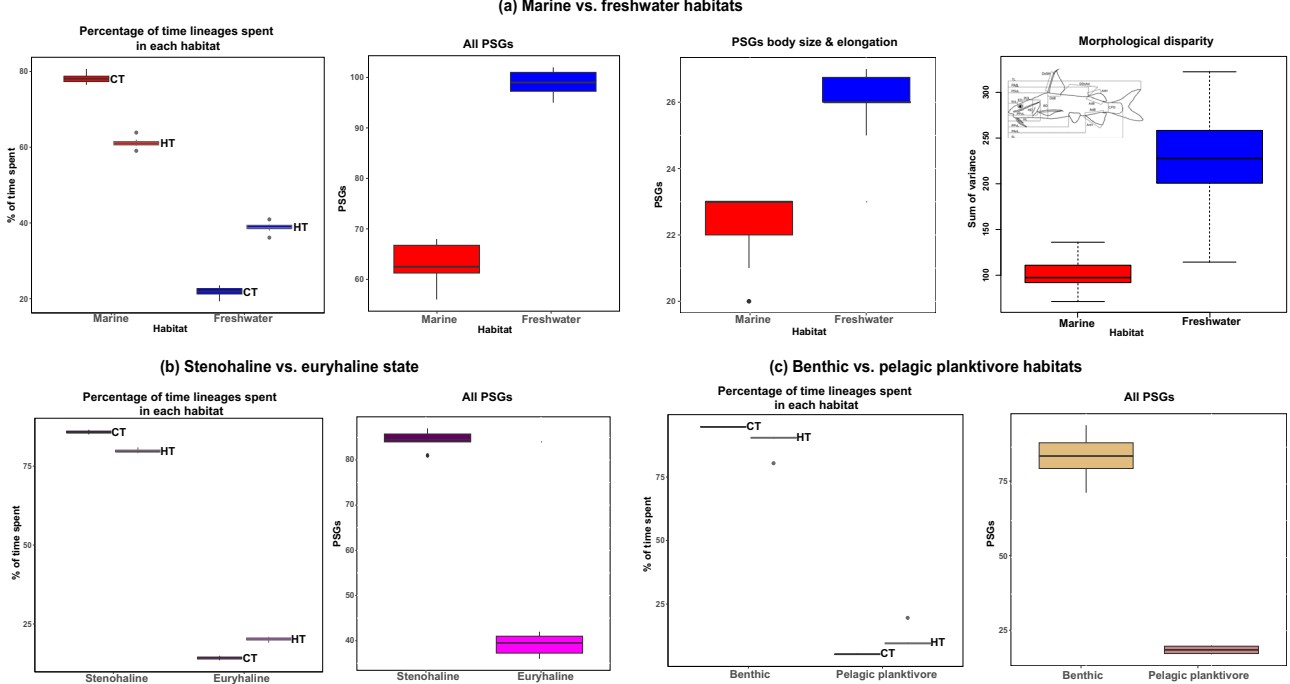

**Fig. 3 | Evolutionary time ariid lineages have spent in each habitat or state, and the number of positively selected genes (PSGs) for each, based on the ten analyzed trees.** Time spent in each habitat is differentiated in two ways: (i) complete tree (CT; black stroke; 119 species), and (ii) sampling restricted to taxa with whole-genome sequences used for HyPhy analyses (HT; gray stroke; 66 species). **a** Ariid lineages spent ~78% of their evolutionary history in the ocean (red) and 22% in rivers (blue) (61% vs. 39%, respectively, considering trees with whole-genome species only). In both scenarios, freshwater lineages exhibit a higher number of genes under positive selection, both in total count and specifically for body size and elongation, compared to marine lineages, aligning with the greater morphological disparity observed in younger freshwater lineages, with a mean disparity of 234.85, compared to 103.1 in their more ancient marine counterparts (refer to

*Supplementary Information*, Fig. S19 for similar analyses of marine-to-freshwater transitions coded after excluding euryhaline species). **b** Euryhaline species (magenta) and (**c**) pelagic planktivores (pink), representing derived (test) states or habitats, have considerably fewer PSGs compared to their ancestral (reference) counterparts (stenohaline, purple; benthic, amaranth). While this coincides with the proportional evolutionary time spent in each habitat (15–20% euryhaline, 5–17% pelagic versus 80–86% stenohaline, 83–95% benthic), these two transitions also involve fewer lineages in the derived state compared to marine-to-freshwater transitions. Therefore, the relative proportions of PSGs are not necessarily comparable across the three different transitions. Boxplots depict the median (center line), interquartile range (box), and range (whiskers). Source data are provided as a Source Data file.

on SIMMAP analyses (see *Supplementary Information*, Supplementary Materials and Methods). These approaches collectively identified 66 shared PSGs (Supplementary Data S11). When comparing ancestral (marine) and derived (freshwater) habitats, HyPhy analyses revealed more genes under positive selection in freshwater lineages (61) despite their younger evolutionary history (22–39% time; see above) compared to the more ancient marine lineages (43 PSGs; 61–78% time; Fig. 3).

The higher number of PSGs in younger freshwater lineages challenges the intuitive expectation that older lineages should harbor more PSGs due to the increased opportunity for beneficial mutations to arise and become fixed over time. This discrepancy likely reflects the complex and dynamic nature of molecular evolution, where factors like fluctuating selective pressures, varying mutation rates, and episodic bursts of adaptation can obscure simple correlations between PSGs and time[56,57]. The observed pattern suggests that the rapid adaptation to novel selective pressures of freshwater environments may have driven accelerated positive selection in these lineages. Importantly, our "control" analyses of salinity tolerance and water column transitions, where older lineages, as expected, exhibit more PSGs (see below), suggest that the observed pattern in freshwater lineages is not solely a result of methodological bias in detecting positive selection in younger lineages. However, it should be noted that these other transitions involve fewer derived lineages, which may lead to reduced statistical power and limit direct comparison of the relative proportions of PSGs across all lineages. Collectively, our results underscore the

importance of considering multiple evolutionary forces, beyond simple time-dependent accumulation when interpreting patterns of positive selection across different lineages and ecological contexts. The individual PSGs identified in freshwater lineages also provide insights into the genetic mechanisms underlying adaptation to this unique environment.

## Patterns of positive selection across freshwater lineages correlate with levels of phenotypic disparity

To identify a possible association between phenotypic disparity and positive selection across marine and freshwater habitats, we first reanalyzed a previously published morphometric dataset for ariids[13], consisting of 28 morphometric and 2 meristic traits compiled from 118 species. We found a significantly greater mean disparity (234.8) in younger freshwater lineages compared to their more ancient marine counterparts (103.1; Fig. 3). In freshwater lineages, 26 (of 61) PSGs are associated with body size and elongation, while in marine lineages, 22 (of 43 PSGs) are linked to these traits (*Supplementary Information*, Figs. S15, S16; Supplementary Note 5). Although the proportion of PSGs associated with these traits is higher in marine lineages, the total number of relevant PSGs is greater in freshwater lineages. This observation seems to align with the higher phenotypic disparity observed in freshwater lineages. These results stand in stark contrast to a recent study[55] that found no significant differences in body size disparity alone between these habitats. For a comparison of phenotypic disparity among radiating and non-radiating freshwater lineages see *Supplementary Information*, Fig. S20.

PSGs in the body shape and size category include genes from the Insulin-like growth factor and transforming growth factor-beta families, also found in studies with cichlid fishes[5,54], along with genes related to development and growth[24] such as *CFAP53, HBP1, LRRK2*, and *USF3* (Fig. 2; *Supplementary Information*, Fig. S17). The category with the second-highest number of PSGs was associated with immune processes[15,58,59], with four in freshwater compared to six in marine lineages. Notable PSGs in this category include genes such as *TNK1*, involved in innate immunity, *IL-11*, associated with immune cell activation, *SGPL1*, linked to immune processes, and *DNAJA2*, involved in developmental stages and stress responses. Other PSGs linked to marine-to-freshwater transitions include osmoregulatory genes[25,59] like aquaporin 7 (*AQP7*) a water channel protein linked to cell volume regulation and sensing, and prolactin (*PRL*), known for its role in regulating chloride cells with a gradual action, vision-related gene[60] like pygopus family PHD finger 2 (*PYGO2*), involved in the development and function of the visual system in fishes, and fin-shape genes[54] attributed to changes in the regulation of fin ray growth like the fibroblast growth factor family (*FGF*) and exostosin-like glycosyltransferase 3 (*EXTL3*).

## Interplay between convergence, parallelism, and historical contingency across marine-to-freshwater transitions

We next investigated the incidence of positive selection during and after transitioning to freshwater habitats (see Methods), identifying 37 PSGs directly involved in the transition (stem lineages, aBSREL), compared to 66 PSGs during and following the habitat shift (foreground lineages, BUSTED-PH). These PSGs were associated with various biological functions, including those related to body size and elongation[5] (e.g., *ATP6V1G1, CDK17, CFAP53*), immune function[15] (e.g., *HIVEP*), fin shape[30] (e.g., *EXTL3, FOXC1A, FGF4*), erythropoiesis[61] (*ADNP2*), and osmoregulation[25] (*PRL*) (Fig. 2, Fig. S16). As noted below, positive selection in *PRL*, rather than the commonly studied aquaporins, claudins, and vasopressin genes[25,62,63], suggests diverse pathways for fish to adapt to freshwater habitats[64,65]. These genes under selection likely contributed to the successful radiation of sea catfishes in Australo-Papuan rivers (see below), and also challenge more general deterministic views regarding marine-to-freshwater transitions across fish diversity. For example, unlike Ishikawa et al.'s[26] study on the *FADS2* gene, our analyses did not identify multiple copies of *FADS2* in freshwater (or marine) lineages, and the sole copy we identified was not under positive selection, despite the importance of *FADS2* in freshwater adaptation identified by that study. This finding suggests a more complex and multifaceted mechanism for adaptation to freshwater environments (Supplementary Note 6).

Site-specific positive selection tests revealed 663 positively selected sites (PSSs), primarily linked to genes involved in cellular processes, biological regulation, development processes, and immune responses. While no clear signal of parallelism (identical amino acid changes) or convergence (different changes) in PSSs was observed across all freshwater clades, a subset of PSSs was shared between multiple freshwater lineages. Specifically, 396 out of 663 PSSs were shared between at least two freshwater clades. Among the 15 notable PSSs (in 9 PSGs) depicted in Fig. S17, we also observed a balance between convergence and parallelism. Six PSSs displayed convergence, indicating that independent lineages arrived at the same amino acid substitutions. This convergence is expected when there are limited ways in which a protein can evolve to enhance fitness under a specific selective pressure. Such constraints can arise from factors such as the need to maintain protein function or a limited number of viable amino acid substitutions at specific sites. For example, convergent evolution has been observed in poeciliids adapting to sulfide springs, where conserved mitochondrial pathways have undergone similar changes to cope with the toxic environment[66]. Additionally, we observed 12 PSSs exhibiting parallel changes, suggesting independent

but similar amino acid substitutions from the same ancestral state. Parallel evolution can occur due to shared mutational biases or similar selective pressures acting on different lineages.

These results highlight the repeated targeting of specific genes or pathways by selection during the recurrent colonization of freshwaters by marine lineages. The observed convergence in PSSs emphasizes the importance of certain genes or pathways in facilitating adaptation to freshwater environments, potentially due to their limited evolutionary potential in the face of specific selective pressures. The presence of parallel changes suggests that similar selective pressures can lead to predictable evolutionary trajectories in different lineages, even if those lineages have diverged significantly. Together, these results underscore the dynamic nature of adaptation and the complex interplay between ecological constraints, mutational biases, and lineage-specific evolutionary history in shaping the genomic basis of freshwater colonization[64,67].

## Selection on osmoregulatory genes involved in salinity tolerance

In the transition from stenohalinity to euryhalinity, positive selection analyses unveiled a higher count of genes under positive selection in stenohaline lineages (67) compared to euryhaline counterparts (33), a pattern that mirrors ariids' evolutionary history, with most of their time spent in stenohaline states (86%; Fig. 3). Notably, six PSGs were unique to euryhaline lineages, highlighting potential adaptations to fluctuating salinities[58,59,63,68]. Among these genes, Kininogen-1 (*KNG1*), which regulates blood pressure and inflammation may play a critical role in environmental adaptation, and Ceruloplasmin (*CP*), which acts as an antioxidant, contributes to iron homeostasis and possessing antimicrobial properties. Purine nucleoside phosphorylase (*PNP*) is linked to immunodeficiency and autoimmunity, while GINS complex subunit 3 (*GINS3*) plays a critical role in DNA replication initiation and replisome progression, maintaining genomic integrity during cellular division. Furthermore, SYNDECAN-3 (*SDC3*) involvement in inflammation and angiogenesis suggests its potential role in immune response and tissue remodeling, beneficial for adapting to euryhaline habitats. Lastly, translocase of inner mitochondrial membrane 29 (*TIMM29*) is indispensable for efficient mitochondrial function amid fluctuating energy demands, which may be higher for euryhaline species.

Only two osmoregulatory genes, *AQP7* and *PRL*, showed evidence of positive selection in this analysis. This could suggest a potential adaptation in ariids, where euryhaline species may already possess genetic variations in osmoregulatory genes that facilitate adaptation to a wide range of salinities. Consequently, other gene categories potentially associated with broader physiological adjustments might experience stronger selective pressures during the transition to euryhalinity[63,69]. Further analyses found fewer PSGs directly involved during the transition to the euryhaline state (28 for aBSREL) compared to those identified during and after the transition (72 for BUSTED-PH). Within these 72 genes, 472 PSSs were identified, primarily involved in cellular processes (31), biological regulation (22), metabolism (13), development (9), immunity (6), and transmembrane transport (2). Of these PSSs, 58 were shared across at least two euryhaline clades. In Fig. S18, we highlight four notable PSSs (in four PSGs), with only one displaying convergence (position 20 of the *SOX21B* gene), while four exhibit parallel changes. This pattern is commonly observed, as close relatives are more likely to share the same ancestral state before independent substitutions occur[70] (e.g., Fig. S18, position 99 of the *ADARB1B* gene).

## No signatures of molecular convergence in gill raker formation among pelagic planktivores

In our analysis of water-column transitions, benthic lineages, which have predominantly occupied this habitat for 95% of their evolutionary

time, contained 58 PSGs (Fig. 3; Supplementary Note 5). Conversely, pelagic planktivores, spending only 5% of the evolutionary history of ariids in this habitat, showed 13 PSGs (Fig. 3; *Supplementary Information*, Fig. S15). Among the PSGs unique to pelagic planktivores, we identified genes crucial for development and physiological processes[3,5,71], including Zinc transporter 10 (*SLC39A10*), essential for fetal definitive hematopoiesis[72], DNA primase small subunit (*PRIM1*), indispensable for accurate DNA replication and genetic material duplication[73], and *PNP*, associated with immune processes[74]. Moreover, our analysis of amino acid substitutions in 75 PSGs identified 353 PSSs associated with water-column transitions, with 14 PSSs shared among at least two pelagic planktivore clades. Figure S18 highlights five PSSs, including two instances of convergence (position 8 of the *FKBP4* gene and position 95 of the *ATP6V1G1* gene), and one instance of parallel changes (positions 108 and 121 of the immune process *LRSAM* gene). During the transition to pelagic habitats, we identified 17 PSGs using aBSREL, while 75 PSGs were identified both during and after the shift in the water column using BUSTED-PH. Transitions from benthic to pelagic habitats often exert selective pressure on specific PSGs that are linked to body depth[5], such as *ATP6V1G1, LRSAM1, RERGLA*, and *USF3*, which can enhance swimming performance and energy efficiency. Additionally, one PSG associated with fin shape (*EXTL3*) can improve swimming efficiency and maneuverability, aiding in prey capture[54]. These adaptations are observed across various fish radiations, including cichlids[5], sticklebacks[27], and whitefishes[3].

Transitions to pelagic habitats often involve morphological adaptations in cranial structures, including gill raker development (which act as a crossflow filter, enhancing prey retention and limiting prey escape[27]). Notably, the *FGF4* gene, known for its role in various biological processes such as early development (mesoderm induction, patterning, gastrulation, fin development) and organogenesis (heart, skeletal, and nervous system development), is under selection in *B. nox*, the ariid species with the highest raker counts (59) and a member of the AU-NG adaptive radiation. *B. nox* exhibits a single substitution at position 218 within *FGF4*, suggesting a possible link between genetic variation in this gene and morphological adaptations in gill raker development. While this does not confirm a direct causal relationship, a previous study by Glazer et al.[27] identified a broad QTL region associated with gill raker number in sticklebacks, of which *FGF4* is one of many candidate genes. Therefore, it is possible that the observed positive selection signal in *FGF4* of *B. nox* could be related to its high gill raker count, highlighting it as a candidate gene for further investigation into the genetic basis of gill raker diversification in ariids. Contrary to our expectations based on a recent study in alpine whitefish[3], where variation within the *EDAR* gene was associated with variation in gill-raker counts, we did not find similar evidence among pelagic-planktivore ariids.

### The adaptive radiation in Australia and New Guinea is characterized by strong selection in the osmoregulatory gene *PRL*

When analyzing positive selection across the three transitions and comparing the AU-NG adaptive radiation with the remaining test clades, we inferred a consistent trend of fewer genes under positive selection in the AU-NG adaptive radiation than in non-radiating clades (Table 1; Supplementary Data S14–S15; Supplementary Note 6). This trend was mostly evident for BUSTED-PH analyses across all transitions (marine-to-freshwater: AU-NG: 41 vs. remaining clades 49; stenohaline-to-euryhaline: AU-NG: 18 vs. remaining clades 69; water column: AU-NG: 10 vs. remaining clades 73). Notably, key functional categories of PSGs for both radiating and non-radiating clades included cellular processes, biological regulation, metabolism, development, response to stimulus, immunity, and transmembrane transport (Figs. S15, S16). Genes related to body size and elongation were a major contributor to PSGs in the AU-NG radiation during water column and salinity tolerance transitions (50% and 44% respectively). However, during marine-

to-freshwater transitions, these genes were more prevalent in the remaining freshwater clades (27% in AU-NG vs. 43% in the remaining test clades). This suggests distinct evolutionary pressures or strategies between the groups. Selection in immune-related genes was also prominent in the AU-NG clade across all transitions (10 % for water-column transitions, 7% for the marine-to-freshwater transitions, and 5% for the salinity-tolerance transitions), highlighting potential adaptations to combat pathogens and challenges in new habitats[15].

As noted above, two osmoregulatory genes (*AQP7* and *PRL*) have been crucial for salinity transitions[25,63,75]. *AQP7* predominantly underwent positive selection in euryhaline species outside the adaptive radiation (BUSTED-PH analyses), whereas *PRL* (aBSREL and MEME analyses) was primarily under selection within the adaptive radiation during the marine-to-freshwater transition (Fig. 4, S18). Notably, three out of the twelve unique PSSs in the AU-NG clade—out of a total of 663 PSSs associated with marine-to-freshwater transitions—were found solely on the *PRL* gene (positions 121, 162-163). Only one (position 121) of these three sites was also under positive selection outside AU-NG, representing a convergent change with a Neotropical freshwater species (*Chinchaysuyoa labiata*). This highlights the significance of *PRL* as a potential osmoregulatory adaptation, with positive selection specifically identified using the stem branches where the marine-to-freshwater transitions occurred. Additionally, PSSs associated with body size and elongation were found on two genes (*DMRT2* and *RERGLA*) across all AU-NG species, suggesting a potential role in their freshwater diversification[5].

### The strength of selection varies across radiating and non-radiating lineages

Our analyses uncovered a predominant pattern of intensified selection in derived (test) habitats or states across the three distinct transitions, relative to their ancestral (reference) counterparts (Fig. S19). While this observation may suggest a prevalence of intensified selection in response to habitat shifts[57], as mentioned earlier, it is also plausible that variations in the duration of time lineages have spent across the different states influence this pattern (Supplementary Note 7). In marine-to-freshwater transitions, we inferred intensified selection on genes potentially involved in sensory perception[76] (*OR131-2*), defense against pathogens[77] (*KAZALD1*), and iron metabolism[78] (*FLVCR2*). Conversely, *DMRT2*, which plays a role in sex determination and development in zebrafish[79], showed relaxation of selection, indicating a potential change in function. Interestingly, genes involved in pH and calcium homeostasis[59] (*ATP6V1G1, TMEM170A*) exhibited contrasting patterns between radiating and non-radiating freshwater lineages, potentially reflecting different environmental pressures across geographies. Euryhaline lineages displayed intensified selection on genes crucial for DNA replication and nucleotide metabolism[73,74] (*PRIM1, PNP*) and development[80] (*LRRK2, NR2C1*), possibly highlighting adaptations for fluctuating salinity. Relaxed selection on *SIGMAR1* suggests that lineages may already have functional osmotic stress management mechanisms[81]. Finally, for water column transitions, pelagic planktivores showed intensified selection on *PNP* and *LRSAM1*, potentially linked to development, immunity, and nervous system adaptations[72,74]. *PNP*, however, also showed contrasting selection patterns between the AU-NG radiation (GUIS) and non-radiating lineages (GURS) potentially suggesting opposing evolutionary forces in pelagic freshwater habitats (AU-NG clade) than pelagic marine environments (two other Neotropical lineages; Fig. 1).

### No statistical associations between transposable elements or other repeat content and ecological transitions

We analyzed repetitive elements in genomes using RepeatMasker and their potential link to habitat occupancy in 61 ariid species (Supplementary Note 8). To explore the potential association between TE/repeat content and habitat occupancy, we employed a phylogenetic

**Table 1 | Number of positively selected genes (PSGs), initially identified from the set of 119 PSGs (out of 2,310 genes tested) using BUSTED-E and analyzed using aBSREL, BUSTED-PH, and RELAX across the three ecological transitions**

| | HABITAT TRANSITION | | | | |
|---|---|---|---|---|---|
| | Marine-to-freshwater transition (coding euryhaline species based on the most probable tip state) | Marine-to-freshwater transition (excluding euryhaline species and reversals to the ancestral marine condition) | Shared genes between the two coding schemes for the marine-to-freshwater transitions | Stenohaline-to-euryhaline transitions | Benthic-to-pelagic planktivore transitions |
| **aBSREL (stem lineages - during transitions to derived habitats)** | | | | | |
| aBSREL | 40 | 39 | 32 | 28 | 17 |
| **BUSTED-PH: ancestral vs derived habitats** | | | | | |
| BUSTED-PH ANCESTRAL HABITAT | 45 | 49 | 43 | 67 | 58 |
| BUSTED-PH DERIVED HABITAT | 67 | 88 | 61 | 33 | 13 |
| **BUSTED-PH: derived habitats AU-NG vs derived habitats non-radiating clades elsewhere** | | | | | |
| BUSTED-PH AU-NG | 64 | 45 | 41 | 18 | 10 |
| BUSTED-PH REMAINING TEST CLADES | 59 | 51 | 49 | 69 | 73 |
| **RELAX: ancestral vs. derived habitats** | | | | | |
| RELATIVE INTENSIFICATION OF SELECTION OF DERIVED COMPARED TO ANCESTRAL HABITATS | 8 | 7 | 3 | 4 | 2 |
| RELATIVE RELAXATION OF SELECTION OF DERIVED COMPARED TO ANCESTRAL HABITATS | 11 | 9 | 4 | 5 | 5 |
| **RELAX: derived habitats AU-NG vs. derived habitats non-radiating clades elsewhere** | | | | | |
| RELAX GUIS AU-NG | 14 | 6 | 4 | 5 | 1 |
| RELAX GUIS REMAINING TEST CLADES | 13 | 9 | 5 | 3 | 1 |
| RELAX GURS AU-NG | 5 | 7 | 4 | 2 | 1 |
| RELAX GURS REMAINING TEST CLADES | 3 | 2 | 1 | 2 | 1 |
| **Habitat Transitions Identified** | | | | | |
| BASED ON COMPLETE TREES | 10-12 | 10-12 | 10-12 | 9 – 11 | 5 – 6 |
| BASED ON PRUNED TREES USED FOR HYPHY ANALYSES | 9 – 10 | 9 – 10 | 9 – 10 | 9 – 11 | 5 – 6 |

Only PSGs detected in at least nine out of ten analyzed trees are counted.

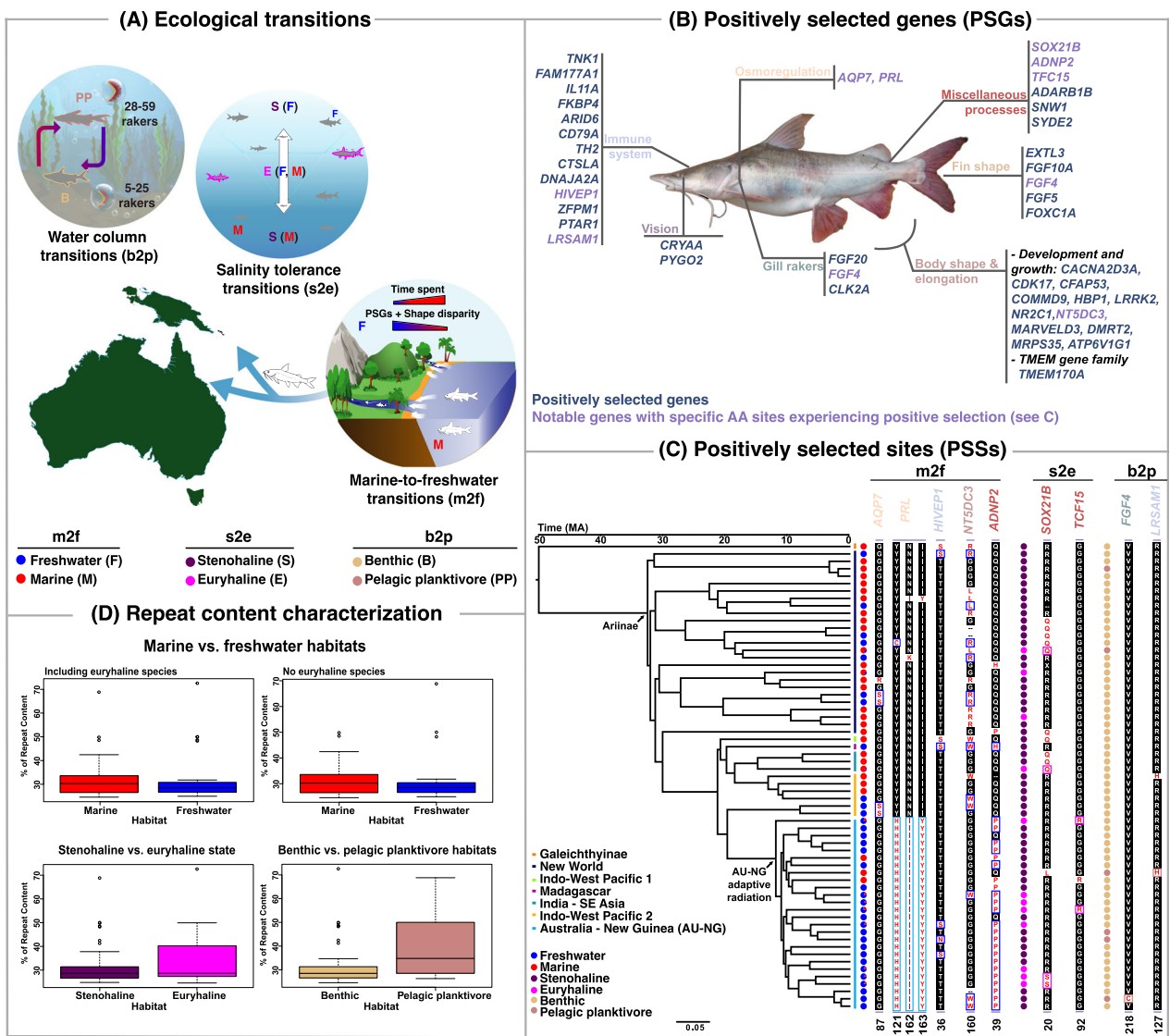

**Fig. 4 | Graphical summary of findings and additional results. a** Three major ecological transitions examined: marine to freshwater, changes in salinity tolerance, and shifts in water column position, emphasizing comparisons between the Australia and New Guinea (AU-NG) adaptive radiation and other clades with derived habitat states worldwide. **b** Noteworthy positively selected genes (PSGs) identified using HyPhy associated with body size and elongation, fin shape, gill rakers, osmoregulation, vision, immune system, and other processes. **c** Convergent and parallel evolution at positively selected sites (PSSs). The phylogeny highlights parallel (same amino acid mutation), and convergent (different amino acid mutation) PSSs identified with MEME. For marine-to-freshwater (m2f) transitions, we illustrate five outstanding genes associated with osmoregulation, immune processes, body size and elongation, and erythropoiesis. Parallel PSSs were found in *AQP7* (position 87) and *HIVEP1* (position 36), while convergent sites were identified in *PRL* (position 121), *NT5DC3* (position 160), and *ADNP2* (position 39). For stenohaline-to-euryhaline (s2e) transitions, results for two genes (*SOX21B*, *TCF15*) are reported, highlighting functions related to metabolic processes and the nervous system. Parallel and convergent PSSs were found in *TCF15* (position 92) and *SOX21B* (position 20), respectively. For benthic-to-pelagic (b2p) planktivore transitions, we show results for two genes (*FGF4*, *LRSAM1*) linked to gill raker formation and immune process. A parallelism was found in *LRSAM1* (position 127), whereas *FGF4* displayed a unique change in *Brustiarius nox*, the ariid species with the highest gill raker count (59). For an extended MEME plot, see Fig. S18. PSSs in derived habitats are denoted by blue boxes for freshwater species, magenta for euryhaline species, and pink for pelagic planktivore species, while changes in the adaptive radiation are indicated in cyan boxes. **d** Repeat content characterization: Comparison of the percentage of repetitive content across marine (red; 34 species), freshwater (blue; 27 species), euryhaline (magenta; 11 species), pelagic planktivore (pink; 6 species), stenohaline (purple; 50 species), and benthic (amaranth; 55 species) lineages, representing different habitat transitions. Despite euryhaline and pelagic planktivore species exhibiting a higher percentage of repetitive elements than stenohaline and benthic lineages, these differences are not statistically significant after FDR correction (see Fig. S21 for additional analyses).

ANOVA approach (*Supplementary Information*, Fig. S21; Supplementary Data S17a). Retroelements were the dominant TE class, followed by DNA transposons (*Supplementary Information*, Fig. S21; Supplementary Data S17a). Marine, euryhaline, and pelagic species exhibited a greater proportion of TEs compared to freshwater, stenohaline, and benthic counterparts (*Supplementary Information*, Supplementary Note 8, Fig. S21, Fig. 4). However, contrary to our expectations, no statistically significant association (FDR corrected *p*-values > 0.05)

between TE content and habitat was found across all three axes (marine-freshwater, stenohaline-euryhaline, water column). It is important to note that, however, due to the variation in the distribution and function of TEs and repeats across different fish species and the general complexity of the genomes of many fish species[82] it can be challenging to confidently identify TEs across the genome and to link TEs and repeat diversity to evolutionary patterns and processes across divergent clades.

## Summary of findings and concluding remarks

Here, we explored the evolutionary history and genomic makeup of sea catfishes, a diverse clade that has undergone numerous, repeated habitat transitions across multiple ecological axes. By investigating major ecological transitions in this clade (including marine-to-freshwater, stenohaline-to-euryhaline, and water column shifts) using comparative genomics in a robust phylogenomic framework, we identified genetic signatures of positive selection in functional categories such as osmoregulation, gill morphology, body shape and size, vision, and other processes. As expected, candidate genes exhibited a higher incidence of PSGs than non-candidate genes across all transitions, suggesting a stronger selective regime in these genes. Most notably, despite ariid lineages spending relatively less time in freshwaters, we found a greater number of PSGs in freshwater lineages compared to their ancient marine counterparts, underscoring the role of ecological opportunity[13] in driving rapid adaptation[11]. Similar findings are observed when looking at the total number of PSGs involved in body size and elongation, aligning with the greater morphological disparity among the relatively younger freshwater lineages.

While convergent evolution was evident in certain genes, suggesting predictable responses to similar selective pressures, the lack of convergence in others, particularly those involved in osmoregulation, emphasizes the importance of historical contingency and unique genetic backgrounds in shaping adaptive outcomes[31,64]. For example, only two osmoregulatory genes, *AQP7* and *PRL*, showed evidence of positive selection during salinity tolerance transitions, with different freshwater lineages having distinct PSGs and PSSs. Other osmoregulatory-related candidate genes known to have undergone adaptive evolution in other marine-to-freshwater ray-finned fish transitions (e.g., *AQP3*, *ATP1A1*, *NOTCH1*) were not under detectable selection in ariids. We also expected other well-known genes associated with habitat adaptation to be under positive selection. However, despite the importance of fatty acid metabolism in adapting to freshwater environments[26], the sole copy of *FADS2* was surprisingly not under positive selection. Although we cannot rule out a lack of statistical power in detecting PSGs or PSSs for many of the candidate genes, this pattern of non-convergence may suggest that marine-to-freshwater transitions can be facilitated by various combinations of genes, indicating a more complex mechanism for ecological adaptation across large evolutionary scales[31,64]. Thus, ancestral marine ariid lineages that repeatedly colonized freshwaters were likely influenced more by ecological opportunity, such as vacant niches[13], rather than by deterministic parallel or convergent shifts in amino acid substitutions.

The adaptive radiation in Australia and New Guinea does not exhibit a notably higher incidence of positive selection relative to freshwater clades from other regions. However, this comparison may be influenced by the greater number of marine lineages undergoing freshwater transitions outside AU-NG. A key osmoregulatory PSG that stands out in the AU-NG clade is *PRL*, which features an elevated incidence of positively selected sites predating the origin of the AU-NG radiation (~13 Ma) and might have facilitated the earlier colonization of marine lineages in this otherwise depauperate freshwater fish region. This early arrival possibly enabled them to radiate without significant competition from other marine-derived fish lineages[13]. A PSS at position 218 of the *FGF4* gene in *Brustiarius nox*, an AU-NG pelagic planktivore with the highest gill raker counts (59) among ariids, suggests a potential link between *FGF4* variation and morphological adaptations in gill raker development[27], possibly in response to dietary preferences and feeding strategies. Finally, although adaptation to new environments has been shown to be accompanied by dramatic changes in repeat and TE content, we did not detect any significant consistent changes in repeat families across transitions.

As one of the first genome-wide studies to examine selection across multiple ecological axes in a non-model fish clade, our research advances our understanding of sea catfish evolution and has broader implications for understanding the genomic basis of adaptation across diverse taxa. It underscores the significant role of colonizing novel ecological regimes in shaping the macroevolutionary trajectories of clades, particularly regarding genomic signatures of selection associated with habitat transitions and patterns of morphological disparity. Future investigations should delve into the potential for adaptive genomic evolution in regulatory or non-coding regions[64], not only in ariids but in fish clades at large.

## Methods

See also *Supplementary Information*, Supplementary Materials and Methods. Due to limits in reference numbers, many supporting references for the discussed genes, as well as the packages used in the various methodological approaches, are provided in the *Supplementary Information*.

### Ethics and inclusion statement

All institutional and/or national guidelines and regulations regarding the collection of specimens were followed. The University of Oklahoma Institutional Animal Care and Use Committee reviewed and approved protocol #2022-0239, entitled "Investigating the factors shaping marine-derived freshwater fish radiations in tropical rivers of Australia and New Guinea." Samples were obtained from the fish tissue collections of various institutions, including the Australian Museum, Louisiana Museum of Natural History, University of Florida, Smithsonian Tropical Research Institute, Muséum national d'Histoire naturelle, INVEMAR, The Academy of Natural Sciences of Drexel University, Museu Nacional do Rio de Janeiro, Museum of Zoology (University of Michigan), Museu de Zoologia da USP, Universidade Federal do Rio Grande do Sul, Museum and Art Gallery of the Northern Territory, and Scripps Institution of Oceanography. Detailed information regarding the specimens used in this study, including associated specimen data and tissue numbers, can be found in Supplementary Data S1.

**Reference genome.** We generated a high-quality reference genome assembly for the lesser salmon catfish from Australia (*Neoarius graeffei*, fNeoGra1, voucher SIO 24-21) in a collaborative effort with the Vertebrate Genomes Project (VGP)[83], using a combination of long-read HiFi PacBio sequences for contig generation, Bionano optical maps and Hi-C for scaffolding, and long read transcriptome sequencing (Isoseq) for genome annotation. BioProject: PRJNA839496 (*Supplementary Information*).

**Assembly of short-read genomes.** We used SPAdes v3.13.1 to assemble the trimmed reads and RagTag v2.1.0 for scaffolding using guided information from the reference genome. To estimate genome statistics such as scaffold N50, size distributions, and assembly size, we used BBMap v38.36. To assess the assemblies' overall completeness, duplication, fragmentation, and relative quality, we ran Benchmarking Universal Single-Copy Orthologues (BUSCO) v5.4.3 using both the Actinopterygii (3,640 genes) and Eukaryota (255 genes) datasets (OrthoDB v10).

**Phylogenomic analyses of exon markers, divergence time estimation and phylogenetic uncertainty in downstream analyses.** In addition to assembling the exon capture data (*Supplementary Information*), we used HMMER v3.2.1 to mine single-copy exons from the 66 genome assemblies. To account for topological and temporal uncertainty[44], we assembled a complete dataset of 1,039 genes and four gene subsets of 265 genes each. The gene subsets overlap only in seven legacy markers, thus maintaining the same set of species. Preliminary analyses using more subsets (12) each with fewer genes (93) resulted in greater topological disparity and thus only analyses based on larger (but fewer) gene subsets were retained (*Supplementary Information*, Fig. S3). We inferred a total of 10 phylogenetic trees using

both concatenation maximum likelihood (ML; RAxML-NG) and multi-species coalescent (ASTRAL-III) based on ML gene trees. Two trees were estimated using the complete expanded matrix, including a 'master tree' based on the RAxML-NG topology; the remaining eight trees were obtained with the four subsets subsampled from this matrix. To account for topological uncertainty (e.g., see ref. [84]), most downstream phylogenetic comparative and genomic analyses used the 10 trees. Some analyses, however, were computationally demanding and thus relied only on the 'master tree' (indicated whenever applicable). We estimated divergence times using the MCMCTree package in PAML v4.9 h, which is suitable for genome-wide datasets and can use as input fixed topologies. For this, we used the 10 tree topologies, thereby accounting for various sources of phylogenetic uncertainty (concatenation vs. coalescent analyses; complete datasets vs. subsets). The MCMCTree analysis was based on 12 primary calibration points (7 fossils and 5 geological events) and 2 secondary points, all derived from two previous studies[13,85]. For topology comparison purposes, we also inferred three additional trees using up to 3551 single-copy BUSCO genes obtained from the 66 whole genomes. See also, *Supplementary Information*, Fig. S8.

**Reconstruction of ancestral habitats.** For the habitat occupancy dataset (Supplementary Data S3), we assigned species into major habitat categories (i.e., marine, euryhaline, freshwater, benthic, pelagic planktivore) by aggregating information from a wide range of sources from the primary literature[13,34]. Using stochastic character mapping analyses (SIMMAP), as implemented in the R package phytools v1.2-0, we conducted three independent analyses for each dataset: (1) transitions between marine and freshwater habitats, where euryhaline species were coded with ambiguity; (2) transitions between stenohalinity to euryhalinity, where marine and freshwater species were coded as stenohaline and those present in both habitats as euryhaline; and (3) transitions along the benthic-pelagic axis in the water column, where pelagic planktivore species were codified based on high gill raker counts as a proxy[13,34,36,51] and diet information (*Supplementary Information*, Supplementary Table S2).

**Candidate and non-candidate genes for PhyloG2P analyses.** We compiled a candidate gene dataset, consisting of 301 genes, by aggregating information from various sources. These sources included gene families associated with body size and elongation[24,29,55], locomotion[29,30], gill-rakers[24,27], trophic morphology[24], osmoregulatory system[52,53], immune genes[58], and vision[24] (Fig. S9). For each of the 301 candidate genes, we retrieved 3-5 ortholog sequences of teleost fishes from the NCBI Orthologs database. We then examined a group of non-candidate genes composed of 101 single-copy genes identified using BUSCO (Actinopterygii OrthoDB) and single-copy orthologues from OrthoFinder (See below for details; *Supplementary Information*). We also obtained a dataset of 3,208 single-copy orthologues from the proteins of eight different Siluriformes species using OrthoFinder (See below for details; *Supplementary Information*).

We employed the hidden Markov model (HMM) approach in HMMER v3.2.1 to identify nucleotide sequences within the assembled genomes for the extraction of 301 target candidate genes. The candidate genes encompassed 167 genes linked to body size and elongation, 18 genes related to fin shape, 14 genes associated with gill-raker counts and anatomy, five genes related to jaw length, 28 genes associated with the osmoregulatory system, two genes related to vision, two genes related to fatty acid metabolism, and 13 genes related to the immune system (Supplementary Data S10). Additionally, we retrieved non-candidate genes, consisting of 3,208 single-copy orthologues, and 432 single-copy orthogroups from the Actinopterygii OrthoDB from BUSCO. Sequence alignments were performed using the reading-frame aware aligner tool MACSE v2.03 and Geneious Prime® v2022.1 to visually inspect the alignments. Only genes meeting the criteria of >70% representation across species, a sequence length >200 bp, and a pairwise similarity of at least 70% were retained. Ultimately, we successfully retained 249 of the targeted 301 candidate genes, and 2,061 non-candidate genes. This includes 1,965 of the 3,208 single-copy orthologs, and 96 single-copy orthologues of the 432 orthogroups with orthologs from the Actinopterygii OrthoDB, totaling 2,310 genes (*Supplementary Information*). Additionally, we used PANTHER v2.2 to acquire gene ontology (GO) data, encompassing GO terms, gene family names, and biological processes linked to candidate genes, SCOs, and BUSCO genes.

**Episodic diversifying selection of candidate and non-candidate genes.** We employed various modules from the HyPhy v2.5.58 package to assess the strength of natural selection using the *dN/dS* metric and to investigate lineage-specific adaptations in marine to freshwater transitions, stenohaline to euryhaline transitions, and water column transitions. These methods included aBSREL, BUSTED-E, BUSTED-PH, RELAX, and MEME. Briefly, aBSREL is a method for identifying specific branches subject to episodic diversifying selection (EDS); MEME is a method for identifying specific codon sites subject to EDS; BUSTED-E is a method to determine whether a gene as a whole is subject to EDS while accounting for alignment errors; RELAX is used to compare selective pressures between two sets of branches in a tree; and BUSTED-PH was developed to test whether or not selection on a subset of branches is associated with a discrete phenotype. We tested the dataset consisting of 2310 genes, which comprised both candidate and non-candidate genes, for signatures of positive selection. For each transition, we designated branches with the 'phenotype' (i.e., habitat) of interest as foreground based on the results of ancestral habitat reconstruction (*Supplementary Information*). We applied an FDR correction to the *p*-values using the p.adjust tool of the R package stats v3.6.2, employing the Benjamini-Hochberg procedure in R with a stringent cut-off of 0.05. We first used BUSTED-E to filter gene sets by identifying regions with potentially aberrant variation patterns that might indicate false positive signals due to sequencing or alignment errors. Initial comparisons between BUSTED-PH, aBSREL, and BUSTED-E revealed notable discrepancies in the number of positively selected genes (PSGs; 84, 189, 26 genes, respectively), implying that the first two methods, lacking accountability for such errors, may overestimate the impact of positive selection (*Supplementary Information*, Fig. S14). We thus selected 119 BUSTED-E genes for downstream analyses across different transition schemes. Our analyses also show that dN/dS methods are sensitive to the phylogenetic tree used (Supplementary Data S15), a concern we addressed by re-running analyses across multiple trees and selecting PSGs consistently identified across them. Results reported above are based on a 9–10 trees cutoff. To identify genes undergoing immediate selection effects during habitat transitions, we utilized aBSREL, targeting the stem lineage of clades featuring the foreground (derived) habitat state. We then used BUSTED-PH to assess whether a gene experienced positive selection on background branches (ancestral states) and multiple foreground branches, including stem and crown lineages, enabling us to identify selection signatures emerging during or after the inferred habitat shifts. Finally, to obtain the coordinates (chromosome position, start, end) of PSGs in the *Neoarius graeffei* genome, we conducted a BLAST search of each PSG using the sequence for the *N. graeffei* gene against the *N. graeffei* genome using the NCBI BLAST Genomes tool.

**Morphological disparity.** We reanalyzed a morphometric and meristic dataset for 118 species of ariids[13], which consists of data on body shape and maximum total length, as well as counts of anal-fin rays and gill rakers. Using the R package dispRity v1.7.0, we quantified morphological disparity using the sum of variance disparity statistics, calculated for different ecological groupings using the first five morphospace axes scores. These analyses were conducted to compare the levels of

disparity between (i) marine and freshwater lineages, and (ii) among the AU-NG adaptive radiation, freshwater lineages in other regions, and marine lineages, in order to explore their association with the number of genes under positive selection for body shape and size across the different clades.

**Transposable elements and repeat content characterization associated with habitat transitions.** For analyses of repeat content characterization, we used RepeatModeler v2.0.4 to construct a de novo transposable element library from the lesser salmon catfish reference. This library was then used to predict transposable elements within each ariid genome using RepeatMasker v4.1.4. All repeat elements were classified into ten categories, including long interspersed nuclear element (LINE), short interspersed nuclear element (SINE), long terminal repeat (LTR), transposons, rolling circle elements, small RNAs, satellites, simple repeats, low complexity repeats, and unclassified. To compare transposable elements and repeat content across habitat transitions, we utilized phylogenetic analysis of variance on master, alternative, and subset trees. Employing the RRPP method with 1000 simulations and post-hoc tests, we assessed the impact of habitat transitions on repeat content, correcting $p$-values using the Benjamini-Hochberg procedure (cutoff: 0.05). To explore the potential association between TE/repeat content and habitat occupancy, we employed a phylogenetic ANOVA approach.

### Reporting summary

Further information on research design is available in the Nature Portfolio Reporting Summary linked to this article.

## Acknowledgements

We thank P. Chakrabarty (LSU Museum of Natural Science) and M. Sabaj (The Academy of Natural Sciences of Drexel University) for providing tissue samples, and A. Marceniuk for valuable comments on ariid systematics, as well as morphometric data and tissue samples. We also thank M. Tan (University of Illinois Urbana-Champaign) for providing exon capture data. Special thanks are due to R. Peterson (The George Washington University), A. Santaquiteria, and U. Rosas (The University of Oklahoma) for their insights on methodology and study design. We acknowledge T. Michael and N. Allsing (Salk Institute) for their assistance with genome annotation. Bioinformatic analyses were performed at the University of Oklahoma Supercomputing Center for Education & Research (OSCER). Funding for Illumina sequencing was provided by Iridian Genomes grant IRGEN_RG_2021-1345: Genomic Studies of Eukaryotic Taxa. This research was supported by National Science Foundation (NSF) grants DEB-1541491 and DEB-2225130 to R.B.R., DEB-2225131 to D.D.B, and DEB-2144325 to D.A; and by a National Institutes of Health grants GM151683 (NIGMS) and HG009299 (NHGRI) to S.K.P.

## Author contributions

R.B.R., M.R.S., R.D.K, D.B. designed the study. R.B.R., A.D., and A.K. collected and curated tissue samples. R.B.R., and M.R.S. organized the taxonomic sampling and extracted the DNA. M.R.S. and S.P. conducted bioinformatic analyses for data assembly. R.B.R collected morphological data. M.R.S and R.B.R conducted ancestral habitat reconstructions. J.M. reference genome DNA isolation. J.B. conducted reference genome PacBio sequencing. B.O.T conducted reference genome IsoSeq sequencing. L.A assembled the reference genome. A.T. curated the reference genome. E.J., G.F., and J.B. supervised VGP team. M.R.S. and R.B.R. estimated divergence times. M.R.S., and D.A. conducted genomic analyses. M.R.S., and R.D.K. conducted repetitive content analyses. M.R.S., S.D.S., and S.K.P. conducted positive selection analyses. M.R.S, R.D.K., and R.B.R. made the figures. R.B.R., M.R.S., and R.D.K. wrote the manuscript; all other authors contributed to and revised the final version of the manuscript.

## Competing interests

The authors declare no competing interests.

## Data availability

Data deposition: All newly sequenced genomes for this study and their raw reads are available from NCBI under the BioProject accession numbers listed in Supplementary Data S1, along with associated specimen information, tissue numbers, and sample locations. The lesser salmon catfish reference genome used is available under RefSeq accession GCF_027579695.1. A FigShare Repository contains alignments, trees and code used for data analysis https://doi.org/10.6084/m9.figshare.25924390. Source data are provided as a Source Data file. Source data are provided with this paper.

## Code availability

HyPhy analysis scripts are available on GitHub: https://github.com/veg/hyphy and https://github.com/stephenshank/catfish. Repetitive content analysis scripts are available on GitHub: https://github.com/RishiDeKayne/Betancur_Lab_Notes/blob/main/02_repeat_characterisation_commands.txt.

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

## Additional information

[1]Department of Biology, The University of Oklahoma, Norman, OK 73019, USA. [2]Department of Ecology and Evolutionary Biology, University of California Santa Cruz, Santa Cruz, CA 95064, USA. [3]Institute for Genomics and Evolutionary Medicine, Temple University, Philadelphia, PA 19122, USA. [4]Iridian Genomes, Silver Spring, MD 20904, USA. [5]School of Natural & Physical Sciences, The University of Papua New Guinea, University 134, National Capital District Port Moresby, Papua New Guinea. [6]Vertebrate Genome Lab, The Rockefeller University, New York, NY 10065, USA. [7]Scripps Institution of Oceanography, University of California San Diego, 8622 Kennel Way, La Jolla, CA 92037, USA. [8]Centre for Tropical Water and Aquatic Ecosystem Research, School of Marine and Tropical Biology, James Cook University, Townsville QLD 4811, Australia. [9]Department of Biological Sciences, Western Michigan University, Kalamazoo, MI 49008, USA. [10]These authors contributed equally: Melissa Rincon-Sandoval, Ricardo Betancur-R. ✉e-mail: rbetancur@ucsd.edu

