## [Peer Review file · Nature Communications]

Ecological diversification of sea catfishes is accompanied by genome-wide signatures of positive selection

Corresponding Author: Professor Ricardo Betancur-R

Version 0:

Reviewer comments:

Reviewer #1

(Remarks to the Author)

This is a relatively straightforward manuscript that investigates genomic signatures of adaptive selection during the ecological diversification sea catfishes (Ariidae). The topic of the article is timely, and the authors present new genomic data that will certainly be of great interest for current and future research studies on fishes. The manuscript is in general well written, and the methods employed are sound. I only have a few comments and points to be addressed, which I indicate below.

- Line 55: Please cite here also the recent paper by Martinez-Redondo et al. (<https://doi.org/10.1111/mec.16854>) that is relevant for repeated/parallel transitions from marine to freshwater in arthropods and several other invertebrate groups.

- Line 73: Please specify the six different gene families mentioned in this sentence.

- Line 338 (and related paragraphs in Discussion and Conclusions about aqp7): Please cite here also the recent paper by Lorente-Martinez et al. (<https://www.mdpi.com/2079-7737/12/6/846>) that is relevant because it reports positive selection in AQP7 (and other aquaporins) in amphibious fishes in their transition from marine to freshwater and terrestrial environments.

- Line 444: Please change "reference" to "background", as it is likely more appropriate in this context (i.e. foreground vs. background).

- Line 593: Please explain why MCMCTree was preferred over (widely used) BEAST.

I hope the authors find my comments helpful.

Reviewer #2

(Remarks to the Author)

This is an interesting manuscript with an impressive dataset, including the long-read genome assembly of one catfish species, plus short-read resequenced genomes of 66 additional species, and additional genetic data from 54 other species. The authors use this dataset to propose a comprehensive and well-supported phylogeny of Ariidae (catfish), to reconstruct the evolutionary history of three major ecological transitions (marine to freshwater, stenohaline to euryhaline, and benthic to pelagic), and to determine how the molecular evolution of genes has been affected by these transitions. They conclude that these transitions have had a major impact on the evolution of catfish genomes. Although I think this is an interesting paper, I have some concerns about the methods and interpretations. Also, I felt that the conclusions mostly restate the results, but do not tell us how this study advances our understanding of how genes/genomes evolve in response to major transitions, or how these results clarify standing questions in gene evolution (e.g., convergence vs. contingency) (but line 538-540).

L310-320: The correlation between patterns of positive selection and levels of phenotypic disparity is very unclear. First, it is not clear how this is a test. Reading the text gives the impression that it is the association between two variables in two groups. There is more morphological disparity in freshwater lineages than in marine lineages. That is clear. But the

association with positively selected genes is not clear. Genes associated with body size and elongation in freshwater lineages are 22 (out of 61 PSGs) and in marine lineages are 20 (out of 43 PSGs). How 22 is (statistically) different from 20, and how this suggests a correlation between PSGs and morphological disparity.

L347: Here, as elsewhere, it is suggested that the number of positively selected genes should correlate with the evolutionary time spent in each of the two environments. It will be useful to make clear why this should be expected. If environments are stable over long periods of time, then there is no reason to expect time to be associated with the number of PSGs. I wonder if this is more related to the independent colonizations of freshwater, each solving similar ecological problems with different genes, regardless of time spent in each environment.

L534-538, and elsewhere: The expectation of convergence in PSS is not clearly stated in the manuscript. In particular, the authors do not provide a clear framework for understanding why convergence in amino acid substitutions is expected. Convergence in amino acid substitutions could be expected in cases of incomplete lineage sorting. For example, there is large standing genetic variation in marine populations of sticklebacks, and therefore cases of genetic convergence are seen because alleles segregating in marine populations are selected in freshwater environments. However, genetic convergence would also be expected if there were few mutations that would increase fitness under selection. For example, poeciliids adapted to sulfide springs show convergent evolution of conserved mitochondrial pathways. There are limited ways to solve the problem of sulfide toxicity, so convergence is expected. I have trouble clearly fitting the catfish work presented into either of these two cases, so it will be important to clearly state why convergence is expected here and what the mechanism might be.

L545, but also 32 and 404-405: The logic behind the interpretation of selection in the PRL gene as a "preadaptation" to the marine-freshwater transition is unclear. PRL shows evidence of positive selection in the AU-NG clade, and there are several PSS under positive selection in this clade. This suggests that this is an ecologically relevant gene. However, it is unclear how this can be considered a "preadaptation". Perhaps the authors could explicitly explain this conclusion.

L549-551: Here, as in other parts of the manuscripts, there is a tendency to overstate the results of models of molecular evolution. The authors add the word "potential" to this sentence, but the standards of the field have grown so much in the last decade that this kind of statement is hard to support. FGF4 is putatively involved in many different processes, and FGF4 morpholinos show phenotypes in very different organs. So the suggestion that FGF4 is involved in the evolution of gill rakers because a species with a high number of gill rakers has a PSS in FGF4 is not justified. The paper cited to justify this conclusion (Glazer et al. 2014 *EvoDevo*) identifies broad QTLs associated with gill raker number, and *fgf4* is one of many candidate genes. Although I have expanded in this particular case, the authors have suggested a few candidate genes based on phenotypic correlations. As this may guide functional analyses, it may be wise to be more explicit about the strength of evidence for associations between phenotypes and PSS.

Figures: Although the figures are nice, the font used is generally very small and many figures are hard to read. For example, the ancestral state reconstructions in Figures 1b-d are very hard to read, especially in d where the colors are not that different. Even when I printed the figure as a full page, it was hard to read. The gene names in Fig. 2 and their color codes are very hard to see, the font is too small. Also in Fig. 2, the circle and hexagon are very hard to distinguish in the figure.

Figure 3: Just to double check, is the variation seen in these box plots the result of using different trees? Or is it variation between the branches being tested?

I cannot find any information on how the various HyPhy schemes (e.g. Fig. S4 and Fig. S5) are tested. How are the models that assume such schemes tested against models that do not assume such schemes?

L227-233: Although when I read it I thought I understood the analyses to determine how much time lineages spend in each of the different states (e.g., marine vs. freshwater), I am now convinced that I do not understand them. What I understand is that the analysis takes the whole lineage and determines what percentage of evolutionary time the "whole" lineage spent in one state or the other. So the sum of both is expected to be 100% (e.g., 78% + 22% in m2f, 85% + 15% in s2e, and 95% + 5% in b2p). But the text is confusing. "...proportional time spent in each habitat by ancestral versus derived lineages". How is the freshwater lineage supposed to spend time in marine environments? In the figure caption it says, "...freshwater species (blue) spent 22% [of their evolutionary history] in rivers". I don't think this makes sense. freshwater species probably evolved in freshwater, especially those in the Au-NG radiation. So to say that each species spent only 22% of its evolutionary time in rivers sounds strange. I think it should be that the Ariid lineage (including all species) spent 78% of its total evolutionary history in marine environments and only 22% in rivers. Or am I missing something?

L306: Which scenarios?

Reviewer #3

(Remarks to the Author)

This is a well-written and interesting paper, and might I even say fun – I like their approach to looking at correlations

between the genome and phenotype across an overlooked lineage (marine catfishes). Marine catfishes aren't the sexist group (I guess scientists and fishermen are alike on this take) but their broad habitat range is notable. I don't think I've read a paper like this yet and it appears to be on the bleeding-edge of what we are doing in systematics, which is using genomics to actually start showing correlation with high-quality genomic/phenotypic/ecological comparisons (causation will come next with CRISPR I suppose). The authors do cite a few similar papers in the Introduction but they are far from common, and this might be the best one. This study looked at an impressive number of genomes (nearly 70) from Ariid catfishes from Australia and New Guinea, not an easy group to get samples from so I appreciate the museum work done here.

The big picture interesting result/generalization based on the correlation of phenotype and genotype was that: "relatively younger freshwater lineages exhibit a higher incidence of positive selection than their more ancient marine counterparts and also display a larger disparity in body shapes, a trend that correlates with a heightened occurrence of positive selection on genes associated with body size and elongation." I think the authors do a nice job showing evidence for that big-picture result; they are exhaustive in their details which is what a paper like this requires.

One issue I have in general with many Vertebrate Genome Project papers is a lack of a link between the genome sequences and the voucher (reference specimens) – I looked through the 52 pages of supplemental materials provided and didn't see a list of GenBank Project #'s for the genomes and the vouchers used for those genomes but did find them in the Excel file linked to the manuscript docs. Thank you for doing that as it is an important part of repeatability and for verifying species identifications in the future. I do hope this file can be somehow be incorporated into the supplemental materials as I think it appears to be a bit buried as currently presented. I do like that the reference genome for *Neoarius graeffei* is given top billing in the manuscript (line 566-570) – why not mention the voucher here? Is it the same one listed in Excel file for the shotgun sequencing? NTM S.15362-002? Or is it the BioSample SAMN28453209 as listed on GenBank under this project. Or are those the same thing? Please clarify both in the manuscript and on GenBank. (And perhaps cite your favorite museum acronym paper or website) in the manuscript with that Excel file.)

This paper really is a modern tour-de-force so kudos to the authors. I even like that the lack of correlation between gill raker count and candidate genes were discussed in a clear fashion that didn't push the author's initial hunch – which was a good one based on the whitefish paper they cite. At the same time, they found surprises like the positive selection in osmoregulatory genes. This is how science should work and unfold.

I'm sorry that due to time constraints I can't do a more in-depth review of the methods, but I wanted to read and review this paper between two trips. I waive anonymity – this is Prosanta Chakrabarty. I notice I am in the acknowledgments for providing some tissue samples, but I have no conflict or reason to not give this impartial review.

The paper is excellent and I look forward to seeing it published.

Version 1:

Reviewer comments:

Reviewer #2

(Remarks to the Author)

I appreciate the reviewers' efforts in refining the manuscript based on my comments and those of the other reviewers. The manuscript is quite solid and represents a valuable contribution to our understanding of how transitions into new ecological niches influence genomic signatures of selection. However, I remain uncertain about the interpretation regarding the association between phenotypic disparity and positive selection. As mentioned in the rebuttal, this analysis is exploratory rather than a formal test. Despite this clarification, line 320 still refers to it as a "test." Moreover, the statement that "more than a third of PSGs in freshwater lineages" are associated with body size needs further clarification. Proportionally, it seems that a larger percentage of PSGs in marine lineages are associated with body size. This observation raises questions about how this finding aligns with the higher phenotypic disparity in freshwater habitats. It appears that there might be a discrepancy or even an opposite trend than what is currently interpreted. In line 563, the manuscript again suggests that freshwater fish have a higher proportion of PSGs associated with body size and elongation. However, when considering the proportion, marine lineages show a higher percentage. If we look at the absolute numbers, I am not convinced that the difference between 22 and 26 is biologically significant. I hope this feedback is helpful and contributes to further strengthening the manuscript.

RESPONSE TO REVIEWER COMMENTS

Reviewer #1 (Remarks to the Author):

This is a relatively straightforward manuscript that investigates genomic signatures of adaptive selection during the ecological diversification sea catfishes (Ariidae). The topic of the article is timely, and the authors present new genomic data that will certainly be of great interest for current and future research studies on fishes. The manuscript is in general well written, and the methods employed are sound. I only have a few comments and points to be addressed, which I indicate below.

Response/ Thanks for all the positive comments!

1.1 - Line 55: Please cite here also the recent paper by Martinez-Redondo et al. (<https://doi.org/10.1111/mec.16854>) that is relevant for repeated/parallel transitions from marine to freshwater in arthropods and several other invertebrate groups.

Response: The citation has now been included:

“Marine-to-freshwater transitions have occurred repeatedly in several animal groups, including annelids¹¹, mollusks¹⁵, arthropods¹⁶, and teleost fishes¹⁷.”

1.2 - Line 73: Please specify the six different gene families mentioned in this sentence.

Response: We have now done that:

“At least six different gene families (i.e., transporter proteins such as aquaporins, ion channels, osmoregulatory hormones and their receptors, metabolic genes, immune genes, and heat shock proteins) have been associated with the ability of marine-derived lineages to successfully colonize and diversify in freshwaters^{5,24,27–30}.”

1.3 - Line 338 (and related paragraphs in Discussion and Conclusions about aqp7): Please cite here also the recent paper by Lorente-Martinez et al. (<https://www.mdpi.com/2079-7737/12/6/846>) that is relevant because it reports positive selection in AQP7 (and other aquaporins) in amphibious fishes in their transition from marine to freshwater and terrestrial environments.

Response: This paper is now cited:

“As noted above, two osmoregulatory genes (AQP7 and PRL) have been crucial for salinity transitions^{25,62,70}.”

1.4 - Line 444: Please change "reference" to "background", as it is likely more appropriate in this context (i.e. foreground vs. background).

Response: To be consistent with the document and figures, we changed "foreground" to "test," as in Figure S5, where test branches correspond to species from the derived habitat and reference branches correspond to species from the ancestral habitat. (We removed the word "background" to avoid confusion.):

“Our analyses uncovered a predominant pattern of intensified selection in derived (test) habitats or states across the three distinct transitions, relative to their ancestral (reference) counterparts (Fig. S19).”

1.5 - Line 593: Please explain why MCMCTree was preferred over (widely used) BEAST.

Response: While BEAST is a widely used and valuable tool, MCMCTree's computational efficiency, model flexibility, suitability for coalescent analysis, and streamlined command-line interface made it the optimal choice for our specific research needs and computational environment.

The choice to use MCMCTree over BEAST was made for several reasons, each aligning with the specific requirements of our study:

1. Computational efficiency: Our dataset consisted of over 1,000 markers, incorporating both exon capture and legacy markers for over 100 species. MCMCTree's efficient algorithms for sampling trees and parameters made it a better fit for handling this computational load compared to BEAST. This efficiency was crucial for timely analysis completion, especially when running multiple iterations on high-performance computing clusters.

2. Coalescent trees: A significant portion of our phylogenetic inferences involved using multi-locus datasets analyzed under the multispecies coalescent model in ASTRAL-III. Because MCMCTree uses pre-estimated topologies as input, we were able to use these trees to estimate divergence times.

3. Ease of use for our specific workflow: While BEAST's graphical user interface (BEAUti) can be user-friendly, our research workflow was heavily reliant on command-line tools and high-performance computing. MCMCTree's command-line interface seamlessly integrated into our existing pipeline, making it a more efficient and practical choice for our large-scale analyses.

Due to word limit constraints, we have now included a brief explanation in the paper: "We estimated divergence times using the MCMCTree package in PAML v4.9h, which is suitable for genome-wide datasets and can use as input fixed topologies. For this, we used the 10 tree topologies, thereby accounting for various sources of phylogenetic uncertainty (concatenation vs. coalescent analyses; complete datasets vs. subsets). The MCMCTree analysis was based on 12 primary calibration points (7 fossils and 5 geological events) and 2 secondary points, all derived from two previous studies^{13,85}."

I hope the authors find my comments helpful.

Response: Thank you for your helpful comments. The suggestion to include references from last year is particularly valuable for our study.

Reviewer #2 (Remarks to the Author):

2.0 This is an interesting manuscript with an impressive dataset, including the long-read genome assembly of one catfish species, plus short-read resequenced genomes of 66 additional species, and additional genetic data from 54 other species. The authors use this dataset to propose a comprehensive and well-supported phylogeny of Ariidae (catfish), to reconstruct the evolutionary history of three major ecological transitions (marine to freshwater, stenohaline to euryhaline, and benthic to pelagic), and to determine how the molecular evolution of genes has been affected by these transitions. They conclude that these transitions have had

a major impact on the evolution of catfish genomes. Although I think this is an interesting paper, I have some concerns about the methods and interpretations. Also, **I felt that the conclusions mostly restate the results, but do not tell us how this study advances our understanding of how genes/genomes evolve in response to major transitions, or how these results clarify standing questions in gene evolution (e.g., convergence vs. contingency)**

Response: Thank you for your comment. We have revised the Conclusions section to better highlight how this study advances our understanding of gene and genome evolution in response to major ecological transitions. We restate the results in this section because the *Results and Discussion* are combined into a single expanded section. This makes it important to distill the key findings for readers. Here are some changes made to broaden the Conclusions:

“Most notably, despite ariid lineages spending relatively less time in freshwaters, we found a greater number of PSGs in freshwater lineages compared to their ancient marine counterparts, underscoring the role of ecological opportunity¹³ in driving rapid adaptation¹¹”.

“While convergent evolution was evident in certain genes, suggesting predictable responses to similar selective pressures, the lack of convergence in others, particularly those involved in osmoregulation, emphasizes the importance of historical contingency and unique genetic backgrounds in shaping adaptive outcomes^{31,60}”.

“As one of the first genome-wide studies to examine selection across multiple ecological axes in a non-model fish clade, our research advances our understanding of sea catfish evolution and has broader implications for understanding the genomic basis of adaptation across diverse taxa. It underscores the significant role of colonizing novel ecological regimes in shaping the macroevolutionary trajectories of clades, particularly regarding genomic signatures of selection associated with habitat transitions and patterns of morphological disparity. Future investigations should delve into the potential for adaptive genomic evolution in regulatory or non-coding regions⁶⁰, not only in ariids but in fish clades at large”.

2.1 - L310-320: The correlation between patterns of positive selection and levels of phenotypic disparity is very unclear. First, it is not clear how this is a test. Reading the text gives the impression that it is the association between two variables in two groups. There is more morphological disparity in freshwater lineages than in marine lineages. That is clear. But the association with positively selected genes is not clear. Genes associated with body size and elongation in freshwater lineages are 22 (out of 61 PSGs) and in marine lineages are 20 (out of 43 PSGs). How 22 is (statistically) different from 20, and how this suggests a correlation between PSGs and morphological disparity.

Response: Thank you for your insightful comment. We acknowledge that the initial phrasing could have been unclear regarding the nature of our analysis and the association between positive selection and morphological disparity. We double-checked the PSGs associated with marine and freshwater lineages and updated the numbers to 26 out of 61 for freshwater and 22 out of 43 for marine lineages, as shown in Figure S20. Our approach was not a formal statistical test but rather an exploratory analysis aiming to investigate a potential connection between positive selection and morphological disparity. We compared the proportion of PSGs associated with body size and elongation in freshwater and marine lineages. This exploratory analysis, along with the observed greater morphological disparity in freshwater lineages, suggests a possible association between PSGs related to body size and elongation and the diversification of freshwater lineages, but highlights the need for further investigation into the potential adaptive

role of these genes in freshwater. We have revised the relevant sections of the manuscript to more accurately reflect this analysis and its implications:

“Genes associated with body size and elongation represented more than a third of PSGs in freshwater lineages, with 26 out of 61 PSGs associated with these traits, compared to 22 out of 43 PSGs in marine lineages (SI appendix, Figs. S15, S16)”.

Supporting Information: “When comparing marine and freshwater lineages, 26 out of 61 PSGs were associated with body size and elongation in freshwater lineages and 22 out of 43 in marine lineages”.

2.2 - L347: Here, as elsewhere, it is suggested that the number of positively selected genes should correlate with the evolutionary time spent in each of the two environments. It will be useful to make clear why this should be expected. If environments are stable over long periods of time, then there is no reason to expect time to be associated with the number of PSGs. I wonder if this is more related to the independent colonizations of freshwater, each solving similar ecological problems with different genes, regardless of time spent in each environment.

Response: There is indeed a complex relationship between the number of PSGs and evolutionary time spent in different environments. While the factors mentioned undoubtedly contribute to the intricate nature of this relationship, our initial expectation that the number of PSGs should correlate with evolutionary time still holds in many cases (see below). Theoretically, the molecular clock posits a linear accumulation of substitutions over time. While the nearly-neutral theory of molecular evolution emphasizes the predominance of neutral mutations, a small fraction of mutations are indeed beneficial and subject to positive selection. If we assume that mutation rates and the proportion of beneficial mutations remain relatively consistent across lineages over time, older lineages should have a greater opportunity to accumulate these beneficial mutations, leading to a higher number of PSGs compared to younger lineages. Although the rate of molecular evolution can vary due to factors like generation time and metabolic rate (Kimura, 1983), and the majority of mutations are deleterious and quickly purged (Wideman, 2019), all else being equal, these factors would affect both older and younger lineages. Therefore, the fundamental principle of increased opportunity for beneficial mutations over time still supports the expectation of a positive correlation between PSGs and evolutionary time. It is, however, possible that dN/dS methods (like HyPhy) may be more likely to differentially identify more positive selection in younger than older lineages (Zhang, 2004), and we acknowledge that our study’s site-based models might not fully capture the heterogeneity in selection over time and across lineages.

However, our results from the stenohaline-to-euryhaline and benthic-to-pelagic transition “controls” align with the expectation that older lineages have more PSGs. This is the case even though the corresponding derived habitat states (i.e., euryhaline, pelagic) are younger than their freshwater counterparts, indicating that the observed patterns do not reflect a potential bias towards the methods’ ability to differentially detect more positive selection in younger lineages. Overall, the results from these transitions “controls” suggest that the age of the lineage still plays a significant role in the accumulation of PSGs, even when considering the complexities of rate heterogeneity and selective pressures.

In the revised manuscript, we clarify this rationale and explicitly acknowledge the potential for heterogeneity in selection and the limitations of our methodological approach. We emphasize that our initial hypothesis is based on simplistic assumptions (“all else being equal”), but we also discuss how deviations from this assumption, such as variations in mutation rates and selective

pressures, effective population size, etc., can influence the observed patterns of positive selection.

Here is the updated text for the manuscript:

“The higher number of PSGs in younger freshwater lineages challenges the intuitive expectation that older lineages should harbor more PSGs due to the increased opportunity for beneficial mutations to arise and become fixed over time. This discrepancy likely reflects the complex and dynamic nature of molecular evolution, where factors like fluctuating selective pressures, varying mutation rates, and episodic bursts of adaptation can obscure simple correlations between PSGs and time^{55,56}. The observed pattern suggests that the rapid adaptation to novel selective pressures of freshwater environments may have driven accelerated positive selection in these lineages. Importantly, our “control” analyses of salinity tolerance and water column transitions, where older lineages, as expected, exhibit more PSGs (see below), suggest that the observed pattern in freshwater lineages is not solely a result of methodological bias in detecting positive selection in younger lineages. However, it should be noted that these other transitions involve fewer derived lineages, which may lead to reduced statistical power and limit direct comparison of the relative proportions of PSGs across all lineages. Collectively, our results underscore the importance of considering multiple evolutionary forces, beyond simple time-dependent accumulation when interpreting patterns of positive selection across different lineages and ecological contexts. The individual PSGs identified in freshwater lineages also provide insights into the genetic mechanisms underlying adaptation to this unique environment.”

2.3 - L534-538, and elsewhere: The expectation of convergence in PSS is not clearly stated in the manuscript. In particular, the authors do not provide a clear framework for understanding why convergence in amino acid substitutions is expected. Convergence in amino acid substitutions could be expected in cases of incomplete lineage sorting. For example, there is large standing genetic variation in marine populations of sticklebacks, and therefore cases of genetic convergence are seen because alleles segregating in marine populations are selected in freshwater environments. However, genetic convergence would also be expected if there were few mutations that would increase fitness under selection. For example, poeciliids adapted to sulfide springs show convergent evolution of conserved mitochondrial pathways. There are limited ways to solve the problem of sulfide toxicity, so convergence is expected. I have trouble clearly fitting the catfish work presented into either of these two cases, so it will be important to clearly state why convergence is expected here and what the mechanism might be.

Response: We appreciate your insightful comment highlighting the importance of clarifying the expectations for convergence in PSSs.

Convergence in amino acid substitutions is indeed expected in lineages adapting to similar environmental challenges due to several factors:

- Limited solutions: As you mentioned, there might be a finite number of amino acid changes that can effectively improve protein function in a given context. This means that different lineages facing the same selective pressure might converge on the same or similar solutions at the molecular level.
- Functional constraints: Proteins have specific functional constraints, limiting the number of viable amino acid substitutions at certain sites without disrupting protein function. This can lead to convergent evolution at these sites in lineages experiencing similar selective pressures.

- Mutational bias: The genetic code and mutation processes inherently favor certain amino acid substitutions over others. This can lead to parallel evolution, where similar changes occur independently in different lineages due to shared mutational biases rather than shared ancestry.
- Environmental filtering: Strong selective pressures in similar environments can eliminate underfit genotypes, leaving only those with convergent or parallel adaptive changes that confer an advantage in that specific environment. This can accelerate the fixation of these changes in multiple lineages, resulting in a pattern of convergent evolution, as observed in poeciliid fishes adapting to sulfide springs (De-Kayne et al., 2024).
- Convergences or parallelisms in PSSs at the same amino acid sites of a gene may further indicate that there are limited ways in which evolution can operate to solve a specific adaptive problem. In contrast, different habitat-derived lineages with genes featuring PSSs at different sites may suggest a greater diversity of potential adaptive solutions.

In our study, the observed convergence in PSSs across different marine-to-freshwater transitions suggests that certain genes or pathways might be repeatedly targeted by positive selection during adaptation to freshwater environments. While these convergences could be due to factors such as limited solutions, functional constraints, or shared mutational biases, it's difficult to isolate a single factor.

We acknowledge that the mechanisms underlying this convergent evolution are not yet fully understood and require further investigation. In the revised manuscript, we elaborate on the expectations for convergence in PSSs, incorporating the concepts of limited solutions, functional constraints, and mutational bias. We also discuss the implications of our findings for understanding the predictability and contingency of evolutionary processes in the context of marine-to-freshwater transitions.

Here is the updated text for the manuscript into a new subsection:

“Interplay between convergence, parallelism and historical contingency across marine-to-freshwater transitions.”

“We next investigated the incidence of positive selection during and after transitioning to freshwater habitats (see Methods), identifying 37 PSGs directly involved in the transition (stem lineages, aBSREL), compared to 66 PSGs during and following the habitat shift (foreground lineages, BUSTED-PH). These PSGs were associated with various biological functions, including those related to body size and elongation⁵ (e.g., *ATP6V1G1*, *CDK17*, *CFAP53*), immune function¹⁵ (e.g., *HIVEP*), fin shape³⁰ (e.g., *EXTL3*, *FOXC1A*, *FGF4*), erythropoiesis⁵⁷ (*ADNP2*), and osmoregulation²⁵ (*PRL*) (Figure 2, Fig. S16). As noted below, positive selection in *PRL*, rather than the commonly studied aquaporins, claudins, and vasopressin genes^{25,65,74}, suggests diverse pathways for fish to adapt to freshwater habitats^{60,75}. These genes under selection likely contributed to the successful radiation of sea catfishes in Australo-Papuan rivers (see below), and also challenge more general deterministic views regarding marine-to-freshwater transitions across fish diversity. For example, unlike Ishikawa *et al.*'s²⁶ study on the *FADS2* gene, our analyses did not identify multiple copies of *FADS2* in freshwater (or marine) lineages, and the sole copy we identified was not under positive selection, despite the importance of *FADS2* in freshwater adaptation identified by that study. This finding suggests a more complex and multifaceted mechanism for adaptation to freshwater environments.

Site-specific positive selection tests revealed 663 positively selected sites (PSSs), primarily linked to genes involved in cellular processes, biological regulation, development

processes, and immune responses. While no clear signal of parallelism (identical amino acid changes) or convergence (different changes) in PSSs was observed across all freshwater clades, a subset of PSSs was shared between multiple freshwater lineages. Specifically, 396 out of 663 PSSs were shared between at least two freshwater clades. Among the 15 notable PSSs (in 9 PSGs) depicted in Fig. S17, we also observed a balance between convergence and parallelism. Six PSSs displayed convergence, indicating that independent lineages arrived at the same amino acid substitutions. This convergence is expected when there are limited ways in which a protein can evolve to enhance fitness under a specific selective pressure. Such constraints can arise from factors such as the need to maintain protein function or a limited number of viable amino acid substitutions at specific sites. For example, convergent evolution has been observed in poeciliids adapting to sulfide springs, where conserved mitochondrial pathways have undergone similar changes to cope with the toxic environment⁵⁸. Additionally, we observed 12 PSSs exhibiting parallel changes, suggesting independent but similar amino acid substitutions from the same ancestral state. Parallel evolution can occur due to shared mutational biases or similar selective pressures acting on different lineages.

These results highlight the repeated targeting of specific genes or pathways by selection during the recurrent colonization of freshwater by marine lineages. The observed convergence in PSSs emphasizes the importance of certain genes or pathways in facilitating adaptation to freshwater environments, potentially due to their limited evolutionary potential in the face of specific selective pressures. The presence of parallel changes suggests that similar selective pressures can lead to predictable evolutionary trajectories in different lineages, even if those lineages have diverged significantly. Together, these results underscore the dynamic nature of adaptation and the complex interplay between ecological constraints, mutational biases, and lineage-specific evolutionary history in shaping the genomic basis of freshwater colonization^{59,60}.

2.4 - L545, but also 32 and 404-405: The logic behind the interpretation of selection in the PRL gene as a "preadaptation" to the marine-freshwater transition is unclear. PRL shows evidence of positive selection in the AU-NG clade, and there are several PSS under positive selection in this clade. This suggests that this is an ecologically relevant gene. However, it is unclear how this can be considered a "preadaptation". Perhaps the authors could explicitly explain this conclusion.

Response: Perhaps the referee missed this around lines 405-434? "Notably, three out of the twelve unique PSSs in the AU-NG clade, out of a total of 663 PSSs associated with marine-to-freshwater transitions, were found on the *PRL* gene (positions 121, 162-163). Only one (position 121) of these three sites was also under positive selection outside AU-NG, representing a convergent change with a Neotropical freshwater species (*Chinchaysuyoia labiata*)."

Of all the PSGs we examined (663 PSSs in total), only 12 PSSs were unique to the AU-NG clade, and 3 out of those 12 were found in PRL alone. Note that this gene was found to be under positive selection using the stem branches selected for aBSREL, an analysis designed to identify PSGs occurring during the marine-to-freshwater transition. But this comment is valid and perhaps a better term is strong selection in PRL, or simply adaptation. We have now rephrased several portions of the text:

Here is the updated text for the manuscript:

"Although positive selection in the AU-NG radiation does not stand out compared to non-radiating lineages, the notable prevalence of selection across the prolactin gene family during the marine-to-freshwater transition suggests that strong osmoregulatory adaptations may have facilitated their earlier colonization of new habitats and subsequent radiation."

“The adaptive radiation in Australia & New Guinea is characterized by strong selection in the osmoregulatory gene *PRL*.”

“This highlights the significance of *PRL* as a potential osmoregulatory adaptation, with positive selection specifically identified using the stem branches where the marine-to-freshwater transitions occurred”

“A key osmoregulatory PSG that stands out in the AU-NG clade is *PRL*, which features an elevated incidence of positively selected sites predating the origin of the AU-NG radiation (~13 Ma) and might have facilitated the earlier colonization of marine lineages in this otherwise depauperate freshwater fish region.”

Supporting Information: “These results suggest that an osmoregulatory adaptation in prolactin in the ancestor of the AU-NG radiation may have facilitated its successful colonization of freshwater habitats in Australia and New Guinea”.

2.5 - L549-551: Here, as in other parts of the manuscripts, there is a tendency to overstate the results of models of molecular evolution. The authors add the word "potential" to this sentence, but the standards of the field have grown so much in the last decade that this kind of statement is hard to support. *FGF4* is putatively involved in many different processes, and *FGF4* morpholinos show phenotypes in very different organs. So the suggestion that *FGF4* is involved in the evolution of gill rakers because a species with a high number of gill rakers has a PSS in *FGF4* is not justified. The paper cited to justify this conclusion (Glazer et al. 2014 *EvoDevo*) identifies broad QTLs associated with gill raker number, and *fgf4* is one of many candidate genes. Although I have expanded in this particular case, the authors have suggested a few candidate genes based on phenotypic correlations. As this may guide functional analyses, it may be wise to be more explicit about the strength of evidence for associations between phenotypes and PSS.

Response: Thank you for your insightful comment. We acknowledge that the original phrasing might have overstated the direct link between *FGF4* positive selection and gill raker evolution. We have revised the relevant section to more accurately reflect the current understanding of *FGF4*'s role and the limitations of our study in establishing a causal relationship. Specifically, we have toned down the language to emphasize the potential association between *FGF4* and gill raker evolution. We now highlight that this gene is involved in various biological processes (e.g., early development such as mesoderm induction, patterning and gastrulation, fin development; organogenesis involved in heart development, skeletal development, nervous system development), and thus the observed positive selection signal might be linked to other phenotypic traits as well. We have also clarified that Glazer et al. (2014) identified a broad QTL region associated with gill raker number, and *FGF4* is just one of many candidate genes within that region. Further functional studies are needed to validate its specific role in gill raker development.

Moreover, we have carefully reviewed other instances in the manuscript where we might have overstated the results of molecular evolution models and have made appropriate revisions to ensure a more cautious and balanced interpretation of our findings. We aim to be more explicit about the strength of evidence for associations between phenotypes and positive selection signals throughout the revised manuscript. We appreciate your feedback, which has helped us refine our interpretation of the data and improve the overall clarity and rigor of the manuscript.

Rephrased text:

“Transitions to pelagic habitats often involve morphological adaptations in cranial structures, including gill raker development (which act as a crossflow filter, enhancing prey retention and limiting prey escape²⁷). Notably, the *FGF4* gene, known for its role in various biological processes such as early development (mesoderm induction, patterning, gastrulation, fin development) and organogenesis (heart, skeletal, and nervous system development), is under selection in *B. nox*, the ariid species with the highest raker counts (59) and a member of the AU-NG adaptive radiation. *B. nox* exhibits a single substitution at position 218 within *FGF4*, suggesting a possible link between genetic variation in this gene and morphological adaptations in gill raker development. While this does not confirm a direct causal relationship, a previous study by Glazer et al.²⁷ identified a broad QTL region associated with gill raker number in sticklebacks, of which *FGF4* is one of many candidate genes. Therefore, it is possible that the observed positive selection signal in *FGF4* of *B. nox* could be related to its high gill raker count, highlighting it as a candidate gene for further investigation into the genetic basis of gill raker diversification in ariids. Contrary to our expectations based on a recent study in alpine whitefish³, where variation within the *EDAR* gene was associated with variation in gill-raker counts, we did not find similar evidence among pelagic-planktivore ariids”

Figures: Although the figures are nice, the font used is generally very small and many figures are hard to read. For example, the ancestral state reconstructions in

2.6 - Figures 1b-d are very hard to read, especially in d where the colors are not that different. Even when I printed the figure as a full page, it was hard to read.

Response. Thank you for your suggestion. We have increased the font size and the size of the circles in the tree. Regarding panel d, the colors are indeed different (amarant and pink); these were chosen from a two-color palette.

2.7 - The gene names in Fig. 2 and their color codes are very hard to see, the font is too small. Also in Fig. 2, the circle and hexagon are very hard to distinguish in the figure.

Response: Thank you for your suggestion. We have increased the font size. However, the color codes for candidate genes remain consistent throughout all figures in the paper, and therefore we cannot change them in this instance. When designing the figure, we explored various geometric shapes (square, triangle, rectangle), and the hexagon proved to be the most distinguishable when used in conjunction with the circle.

2.8 - Figure 3: Just to double check, is the variation seen in these box plots the result of using different trees? Or is it variation between the branches being tested?

Response. That is correct. As shown in Figure 3 and Table S11 (Supplementary Information, supplementary spreadsheet), we present the time spent by ariid lineages in each state on the tree. To avoid misunderstandings, we have clarified in the figure caption that this information is based on the ten analyzed trees.

Rephrased figure caption: “Figure 3. Evolutionary time ariid lineages have spent in each habitat or state, and the number of positively selected genes (PSGs) for each, based on the ten analyzed trees”.

2.9 - I cannot find any information on how the various HyPhy schemes (e.g. Fig. S4 and Fig. S5) are tested. How are the models that assume such schemes tested against models that do not assume such schemes?

Response: All the models used in this study indeed employed the same scheme. In the Supplementary Information (Figure S4), we detailed how we determined the branches of interest (foreground) for each ecological transition based on ancestral habitat reconstruction results. We consistently defined the derived habitats as foreground (freshwater for marine-to-freshwater transitions, euryhaline for salinity tolerance transitions, and pelagic planktivores for water-column transitions) across all HyPhy analyses. By focusing on derived lineages, we aimed to identify genes and pathways that may be specifically associated with adaptation to these novel environments.

In Figure S5, we illustrate the test branches (corresponding to species from the derived habitat) and reference branches (corresponding to species from the ancestral habitat) used for the comparison between ancestral and derived lineages.

2.10 - L227-233: Although when I read it I thought I understood the analyses to determine how much time lineages spend in each of the different states (e.g., marine vs. freshwater), I am now convinced that I do not understand them. What I understand is that the analysis takes the whole lineage and determines what percentage of evolutionary time the "whole" lineage spent in one state or the other. So the sum of both is expected to be 100% (e.g., 78% + 22% in m2f, 85% + 15% in s2e, and 95% + 5% in b2p). But the text is confusing. "...proportional time spent in each habitat by ancestral versus derived lineages". How is the freshwater lineage supposed to spend time in marine environments? In the figure caption it says, "...freshwater species (blue) spent 22% [of their evolutionary history] in rivers". I don't think this makes sense. freshwater species probably evolved in freshwater, especially those in the Au-NG radiation. So to say that each species spent only 22% of its evolutionary time in rivers sounds strange. I think it should be that the Ariid lineage (including all species) spent 78% of its total evolutionary history in marine environments and only 22% in rivers. Or am I missing something?

Response: This is a good point. We have now rephrased this throughout the paper to reflect that it's ariid lineages that have spent 78% of their time in the ocean and 22% of their time in rivers. Here are some examples:

"Ariid lineages spent approximately 78% of their evolutionary history in the ocean (red) and 22% in rivers (blue) (61% vs. 39%, respectively, considering trees with whole-genome species only)"

"Conversely, pelagic planktivores, spending only 5% of the evolutionary history of ariids in this habitat, showed 13 PSGs (Fig. 3; *SI Appendix*, Fig. S15)."

2.11 - L306: Which scenarios?

Response: The scenarios indeed refer to a complete tree and a subsampling restricted to taxa with whole-genome sequences used for HyPhy analyses, as specified in the manuscript.

"Figure 3. Evolutionary time ariid lineages have spent in each habitat or state, and the number of positively selected genes (PSGs) for each, based on the ten analyzed trees. Time spent in each habitat is differentiated in two ways: (i) complete tree (CT; black stroke), and (ii) sampling restricted to taxa with whole-genome sequences used for HyPhy analyses (HT; gray stroke)".

Reviewer #3 (Remarks to the Author):

This is a well-written and interesting paper, and might I even say fun – I like their approach to looking at correlations between the genome and phenotype across an overlooked lineage

(marine catfishes). Marine catfishes aren't the sexist group (I guess scientists and fishermen are alike on this take) but their broad habitat range is notable. I don't think I've read a paper like this yet and it appears to be on the bleeding-edge of what we are doing in systematics, which is using genomics to actually start showing correlation with high-quality genomic/phenotypic/ecological comparisons (causation will come next with CRISPR I suppose). The authors do cite a few similar papers in the Introduction but they are far from common, and this might be the best one. This study looked at an impressive number of genomes (nearly 70) from Ariid catfishes from Australia and New Guinea, not an easy group to get samples from so I appreciate the museum work done here.

The big picture interesting result/generalization based on the correlation of phenotype and genotype was that: "relatively younger freshwater lineages exhibit a higher incidence of positive selection than their more ancient marine counterparts and also display a larger disparity in body shapes, a trend that correlates with a heightened occurrence of positive selection on genes associated with body size and elongation." I think the authors do a nice job showing evidence for that big-picture result; they are exhaustive in their details which is what a paper like this requires.

One issue I have in general with many Vertebrate Genome Project papers is a lack of a link between the genome sequences and the voucher (reference specimens) – I looked through the 52 pages of supplemental materials provided and didn't see a list of GenBank Project #'s for the genomes and the vouchers used for those genomes but did find them in the Excel file linked to the manuscript docs. Thank you for doing that as it is an important part of repeatability and for verifying species identifications in the future.

3.1 - I do hope this file can be somehow be incorporated into the supplemental materials as I think it appears to be a bit buried as currently presented.

Response: The spreadsheets will be made available alongside the supplementary material. We believe this info. is more accessible and easier to navigate in a spreadsheet format compared to including it as large, unwieldy tables in the Supporting Information. We have also included a sentence in the Additional Information section of the manuscript: "Raw sequencing reads are available in the NCBI Sequence Read Archive under the BioProject numbers listed in Table S1 (Supplementary Information), along with associated specimen information, tissue numbers, and sample locations."

3.2 - I do like that the reference genome for *Neoarius graeffei* is given top billing in the manuscript (line 566-570) – why not mention the voucher here? Is it the same one listed in Excel file for the shotgun sequencing? NTM S.15362-002? Or is it the BioSample SAMN28453209 as listed on GenBank under this project. Or are those the same thing? Please clarify both in the manuscript and on GenBank. (And perhaps cite your favorite museum acronym paper or website) in the manuscript with that Excel file.)

Response: Thank you for your suggestion. The information you found in the Supplementary Information pertains to a different individual that was sequenced for the whole shotgun genome. We have now added a new row to include the information associated with the individual used for the chromosome-level genome, and this is also specified in the manuscript.

"We generated a high-quality reference genome assembly for the lesser salmon catfish from Australia (*Neoarius graeffei*, fNeoGra1, voucher SIO 24-21) in a collaborative effort with the Vertebrate Genomes Project (VGP)⁷⁹".

This paper really is a modern tour-de-force so kudos to the authors. I even like that the lack of correlation between gill raker count and candidate genes were discussed in a clear fashion that didn't push the author's initial hunch – which was a good one based on the whitefish paper they cite. At the same time, they found surprises like the positive selection in osmoregulatory genes. This is how science should work and unfold.

I'm sorry that due to time constraints I can't do a more in-depth review of the methods, but I wanted to read and review this paper between two trips. I waive anonymity – this is Prosanta Chakrabarty. I notice I am in the acknowledgments for providing some tissue samples, but I have no conflict or reason to not give this impartial review.

The paper is excellent and I look forward to seeing it published.

Response: Prosanta, thank you so much for your kind comments. It's very gratifying to receive positive feedback on a manuscript we've put so much work and effort into. This was 4.5 years of work!

RESPONSE TO REVIEWERS' COMMENTS

2.1 I appreciate the reviewers' efforts in refining the manuscript based on my comments and those of the other reviewers. The manuscript is quite solid and represents a valuable contribution to our understanding of how transitions into new ecological niches influence genomic signatures of selection.

R/ Thanks for the positive feedback!

2.2. However, I remain uncertain about the interpretation regarding the association between phenotypic disparity and positive selection. As mentioned in the rebuttal, this analysis is exploratory rather than a formal test. Despite this clarification, line 320 still refers to it as a "test."

R/ Thanks for pointing this out. We have now rephrased/toned down the sentence. See response to comment 2.3 below.

2.3. Moreover, the statement that "more than a third of PSGs in freshwater lineages" are associated with body size needs further clarification. Proportionally, it seems that a larger percentage of PSGs in marine lineages are associated with body size. This observation raises questions about how this finding aligns with the higher phenotypic disparity in freshwater habitats. It appears that there might be a discrepancy or even an opposite trend than what is currently interpreted.

R/ Thank you for identifying this issue. The focus should indeed be on the absolute number of PSGs rather than the proportion. We have revised the paragraph as follows:

"To identify a possible association between phenotypic disparity and positive selection across marine and freshwater habitats, we first reanalyzed a previously published morphometric dataset for ariids¹³, consisting of 28 morphometric and 2 meristic traits compiled from 118 species. We found a significantly greater mean disparity (234.8) in younger freshwater lineages compared to their more ancient marine counterparts (103.1; Figure 3). In freshwater lineages, 26 (of 61) PSGs are associated with body size and elongation, while in marine lineages, 22 (of 43 PSGs) are linked to these traits (*SI appendix*, Figs. S15, S16). Although the proportion of PSGs associated with these traits is higher in marine lineages, the total number of relevant PSGs is greater in freshwater lineages. This observation seems to align with the higher phenotypic disparity observed in freshwater lineages."

2.4. In line 563, the manuscript again suggests that freshwater fish have a higher proportion of PSGs associated with body size and elongation. However, when considering the proportion, marine lineages show a higher percentage. If we look at the absolute numbers, I am not convinced that the difference between 22 and 26 is biologically significant. I hope this feedback is helpful and contributes to further strengthening the manuscript.

R/ Thanks for pointing this out. This sentence has also been rephrased to clarify this apparent contradiction: "Similar findings are observed when looking at the total number of PSGs involved in body size and elongation, aligning with the greater morphological disparity among the relatively younger freshwater lineages."